

# Regional co-variability of spatial and temporal soil moisture - precipitation coupling in North Africa: an observational perspective.

Irina Y. Petrova[1*], Chiel C. van Heerwaarden[2], Cathy Hohenegger[1], and Françoise Guichard[3]

[1]Max Planck Institute for Meteorology, Hamburg, Germany
[*]Current affiliation: Laboratory of Hydrology and Water Management, Ghent University, Ghent, Belgium
[2]Meteorology and Air Quality Group, Wageningen University, Wageningen, The Netherlands
[3]CRNM-GAME, CNRS-Météo France, Toulouse, France

*Correspondence to:* Irina Y. Petrova (irina.petrova@ugent.be)

**Abstract.** The magnitude and sign of soil moisture - precipitation coupling (SMPC) is investigated using a probability-based approach and 10 years of daily microwave satellite data across North Africa at 1° horizontal resolution. Specifically, the co-existence and co-variability of spatial (i.e. using soil moisture gradients) and temporal (i.e. using soil moisture anomaly) soil moisture effects on afternoon rainfall is studied at 100 km scale. The analysis shows that in the semi-arid environment of the

Sahel, the negative spatial and the negative temporal coupling relationships do not only co-exist, but are also dependent of one another. Hence, if afternoon rain falls over temporally drier soils, it is likely to be surrounded by a wetter environment. Two regions are identified as SMPC "hot spots". These are the south-western part of the domain (7 - 15° N, 10° W - 7° E) with the most robust negative SMPC signal, and the South Sudanian region (5 - 13° N, 24 - 34° E). The sign and significance of the coupling in the latter region is found to be largely modulated by the presence of wetlands and is susceptible to the amount of

long-lived propagating convective systems. The presence of wetlands and irrigated land areas is found to account for about 30 % of strong and significant spatial SMPC in North African domain. This study provides the first insight into regional variability of SMPC in North Africa, and supports potential relevance of mechanisms associated with enhanced sensible heat flux and meso-scale variability in surface soil moisture for deep convection development.

## 1    Introduction

Soil moisture can affect the state of the lower atmosphere through its impact on evapotranspiration and surface energy flux partitioning (e.g., Eltahir, 1998; Klüpfel et al., 2011). Especially in the "hot spots" of soil moisture - precipitation coupling (SMPC), like the semi-arid Sahel (Koster et al., 2004; Taylor et al., 2012; Miralles et al., 2012), soil moisture exerts strong control on evapotranspiration (e.g., Timouk et al., 2009; Dirmeyer, 2011; Lohou et al., 2014), influencing the development of the daytime planetary boundary layer (BL), and hence convective initiation and precipitation variability. Most of the physical

understanding on how soil moisture could alter BL properties and affect development of convection comes from 1 to 3-D model analyses (Seneviratne et al., 2010; Nicholson, 2015; Rochetin et al., 2017). Observational evidence of the SMPC largely relies on the measurements of recent field campaigns (like African HAPEX and AMMA: Goutorbe et al., 1994; Redelsperger et al., 2006) and hence, is often limited to a short spatio-temporal scale. Such observational analyses present a unique evidence





of environmental conditions preceding convection development (e.g., Lothon et al., 2011) and can be further used as a test-ground to evaluate and improve the physical parameterizations of models (e.g., Couvreux et al., 2013). Both, observational and modelling studies agree reasonably well on the effect of soil moisture availability and heterogeneity on the lower atmospheric stability (e.g., Kohler et al., 2010) and convective initiation at meso-scales (e.g., Taylor, 2010; Birch et al., 2013). However,

the impact of soil moisture on convective precipitation remains more uncertain. At meso-scale, there is a disagreement in the sign of the SMPC between models which use parameterizations of deep convection and observations (Hohenegger et al., 2009; Taylor et al., 2012, 2013). Recent satellite-based analysis has demonstrated that the choice of soil moisture parameter itself (temporal anomaly or spatial gradient) and related differences in physical mechanisms have a direct effect on the resulting sign of the observed SMPC on 100 km scales (Guillod et al., 2015). Our study addresses the question of co-existence of spatial (i.e.

using spatial soil moisture gradient) and temporal (i.e. using soil moisture anomaly) SMPC in the region of the African Sahel at 1° horizontal resolution. To investigate spatio-temporal variability of observed SMPC relationships, we use 10-year satellite records of daily soil moisture from the AMSR-E and 3-hourly TMPA precipitation product.

Both modelling and observational studies reported the possibility of negative as well as positive SMPC (Nicholson, 2015). Spatial gradients in soil moisture can affect BL state and convection initiation through thermally-induced meso-scale circula-

tions recognized on 10 to 100 km scales (Taylor and Ellis, 2006; Taylor et al., 2007). In association with this mechanism and under favourable thermodynamic conditions, convection is likely to be initiated over spatially drier soils, indicating a negative SMPC (Taylor et al., 2011; Garcia-Carreras et al., 2011). However, whether the further development and propagation of moist convection will occur over drier or wetter soils remains subtle. The modelling study of Froidevaux et al. (2014) suggested that negative SMPC is possible under very weak surface wind conditions, and is associated to stationarity of convective systems

once initiated. The opposite sign is expected under a stronger horizontal advection, which will support propagation of the developing moist convection downwind, i.e. from dry to wet soils, and its further amplification over wetter areas. Another important factor is related to the life-cycle of meso-convective systems (MCS) and thus their organization in space and time (Mathon et al., 2002). Small-scale convective systems are expected to be particularly sensitive to surface moisture variability and will propagate preferentially towards spatially drier soil, bounded by a wetter surrounding (Taylor and Ellis, 2006). Alternatively,

larger organized systems have been found to evolve towards wetter soils - areas of increased latent heat flux, convective available potential energy (CAPE) and moist static energy (MSE) (Taylor and Lebel, 1998; Clark et al., 2003). Hence, these systems are expected to be more sensitive to soil moisture availability.

The impact of temporal anomalies of soil moisture on the atmospheric BL and moist convection is largely governed by thermodynamical processes, and may likewise result in a coupling of both signs. Wet soils are expected to lead to an increase

of boundary layer MSE or similarly equivalent potential temperature, through a decreased boundary layer height (BLH) and subsequent less vigorous entrainment (e.g. Eltahir, 1998; Alonge et al., 2007). The enhanced MSE over wet soils is favourable to convective rainfall formation. Dry soils, on the contrary, are associated with a reduced MSE and thus provide lower potential for convection development and may even suppress existing MCS (Gantner and Kalthoff, 2010) or deviate its propagation direction (Wolters et al., 2010). However, modelling and observational evidence indicate that both dry and wet soils, can favour

moist convection, depending on the morning stratification of the lower atmosphere (Findell and Eltahir, 2003a, b) into which





the daytime convective BL is growing (Ek and Mahrt, 1994; van Heerwaarden and Guerau de Arellano, 2008; Couvreux et al., 2012).

The relevance of meso-scale spatial heterogeneity of soil moisture in favouring moist convection over wet and dry temporal soil moisture anomaly was demonstrated by e.g. Clark et al. (2003) and Taylor and Ellis (2006) respectively. However, until

recently no attempts were made to directly compare the temporal and spatial aspects. The first comparison of the spatial and temporal effects of soil moisture on precipitation was presented by the study of Guillod et al. (2015) (hereafter, G15) at 5° horizontal resolution. Applying the probability-based approach of Taylor et al. (2012) (hereafter, T12) to 10-years of global satellite-based soil moisture and precipitation data, they demonstrated that a negative spatial (rain over spatially drier soils) and a positive temporal (rain over temporally wetter soils) SMPC dominate over most of the globe and do not exclude one

another. G15 suggested that the two effects might be interconnected through spatial coupling mechanisms, in which adjacent precipitation would provide required moisture to enhance convection development over spatially but not temporally drier soil. Using multiple data sets, G15 showed that the signal is robust across different input data sets. However, in a few regions, including the Sahel in Africa, an opposite temporal relationship was revealed: spatially and temporally negative coupling was found to co-exist in opposition to the global relationship.

In this study, we further explore spatial and temporal SMPC as well as their co-existence in North African region using the T12 method at a higher $1 \times 1°$ horizontal resolution. Furthermore, we provide insight into regional co-variability of the spatial and temporal effects on afternoon rainfall. The analysis is realized following two main steps:

1. First we focus on identification of the factors that influence the magnitude and variability of the spatial SMPC measure. By doing so we address the question: which physical processes likely underlie the observed spatial SMPC relationship
if any?

2. Then, we analyze the link between the spatial and temporal effects of soil moisture on precipitation by answering the two following questions: are spatial and temporal negative coupling relationships independent, and if not, how do they inter-relate?

We reproduce and apply the probability-based approach of T12 to 10-years of daily AMSR-E soil moisture and 3-hourly TMPA

precipitation records. In contrast to the previous studies, we estimate the temporal and spatial coupling effects at a higher $1 \times 1°$ horizontal resolution, which reveals previously hidden finer-scale effects of land cover features on the SMPC relationship.

The first part of the study includes an analysis of the regional variability and robustness of the observed spatial SMPC distribution at the highest 1° grid. Specifically, the sensitivity of the spatial SMPC relationship to smaller horizontal scales is tested, i.e. from the original 5° to 2.5° and 1° (Sections 4.1-4.2). Identification of the factors relevant for the observed spatial

SMPC distribution includes a sensitivity analysis of the spatial coupling measure to the presence of soil moisture parameter extremes (Section 4.3) and to the MCS life cycle (Section 4.4). The link between the temporal and spatial SMPC is assessed with a correlation analysis (Section 5.1). Section 5.2 discusses the reasons behind the opposite sign of the temporal coupling identified in the North African region as compared to the dominantly positive relationship identified in G15.



## 2 Domain and Data

### 2.1 Study domain

We focus our analyses on the North African region (5 - 20° N, 20° W - 40° E) (Fig. 1, inset rectangular) during summertime (JJAS). This region has been repeatedly pointed out as a "hot spot" of land-atmosphere interactions (Dirmeyer, 2011; Miralles et al., 2012; Gallego-Elvira and Taylor, 2016), and one of the most vulnerable regions with respect to climate change (Dirmeyer et al., 2012; Dirmeyer and Wang, 2014). A major feature affecting the Sahelian climate is the West African Monsoon (Janicot and Thorncroft, 2008), which is associated with a high precipitation variability (Nicholson, 2013). During the monsoon, soil moisture fluctuations are strongly influenced by precipitation at a large range of spatial and temporal scales. Atmospheric and surface fields display strong meridional gradients between 10° N and 20° N (Figure 1, zonal plot), i.e. on the northern flank of the mean position of JJAS rain belt, also referred to as the Inter Tropical Convergence Zone (ITCZ). Wind convergence at the surface is observed further north, around 18 - 20° N, along the Inter Tropical Discontinuity (ITD), where the cool and moist monsoon flow meets hot and dry Saharan air. Associated with the meridional heat gradient, the monsoon circulation and related large-scale structures like the African Easterly Jet (AEJ), as well as synoptic disturbances like the African Easterly Waves (AEWs) largely modulate convection activity over the region (Duvel, 1990; Mohr and Thorncroft, 2006). Additionally, evidence supporting a significant role of the surface state in the triggering of deep precipitating convection is steadily growing (Nicholson, 2015).

Middle July to August conditions may be less favourable for a strong surface influence on convection. Compared to the drier early and late monsoon months of June and September, the wetter period - from July through August - is characterized by a typically lower level of free convection (LFC) (Guichard et al., 2009; Taylor et al., 2011) and less pronounced spatial contrast between fluxes due to more dense vegetation (Kohler et al., 2010; Lohou et al., 2014). In our study, the role of the monsoon dynamics is not directly addressed to preserve maximum sample size for the sake of statistical significance.

We intentionally extend our analysis further eastwards. Despite the inherent zonal symmetry of surface and atmospheric parameters (as in precipitation in Fig. 1), considerable differences exist in rainfall and large-scale circulation regimes between East and West. Distinct orography, intensity of surface and upper level jets and wave disturbances are likely to bring dissimilarities into the sensitivity of convection to the surface state between the two regions. The eastern African domain can also remotely influence convection in the western part of the region via the genesis of westward propagating AEWs (Laing and Carbone, 2008) and long-lived MCSs (Laing et al., 2012). Yet, notably few studies investigated land-atmosphere interactions in eastern Sahel.

### 2.2 AMSR-E soil moisture

Soil moisture (SM) data from the Advanced Microwave Scanning Radiometer - Earth Observing System (AMSR-E, Jun 2002 - Oct 2011) is analyzed in this study. The AMSR-E unit is carried on board of the polar orbiting AQUA satellite, measuring brightness temperatures in 12 channels, at 6 different frequencies (6.9 - 89 GHz) (Njoku and Jackson, 2003). Soil moisture derived from the lowest C-band frequency of 6.9 GHz is used here, as lower frequencies experience less signal contamination



from vegetation and surface roughness, and are able to receive emission information from deeper soil layers (still few centimeters, Owe et al., 2008). The AQUA orbit is sun-synchronous with typically one overpass per pixel per day at either 13:30 or 01:30 local solar time (LST). In order to capture the surface moisture state shortly before afternoon convection activity, only data of ascending day orbit, i.e. 13:30 LST is used here. It is important to note, that the day overpass is prone to higher biases

compared to the night overpass, because of the greater temperature differences between surface and canopy involved in the physics algorithm (Njoku and Jackson, 2003).

We utilize the Level 3 estimates of AMSR-E soil moisture derived with the Land Surface Parameter Model (LPRM, Owe et al., 2008) for JJAS 2002 - 2011. The product is available at $0.25° \times 0.25°$ spatial resolution. The LPRM is not valid for dense vegetation and water bodies. Therefore pixels with an optical depth $> 0.8$ are excluded. Water body and soil moisture quality

masks were adopted from the materials of T12. Accordingly, pixels containing more than 5 % water are excluded, using water body classification of the 1 km Global Land Cover 2000 data set [Available online at http://forobs.jrc.ec.europa.eu/products/ glc2000/products.php]. Application of the soil moisture quality mask, based on the correlation analysis between precipitation and soil moisture data sets, is intended to reduce the number of pixels covered with wetlands (for details see suppl. in T12).

Many days (40 - 50 %) do not contain soil moisture information due to satellite revisit times. Over a given longitude per

day the number of overpasses below 40° N do not exceed one with occasionally daily or every third day sampling (see Fig. 1 in Njoku and Jackson, 2003). The latter significantly reduces the size of the collected rainfall event sample available for the analyses.

The AMSR-E instrument is chosen because it documents a relatively long period and performs better than ASCAT (Dorigo et al., 2010) over sparsely vegetated and deserted areas. The AMSR-E product also proved to be accurate at the precipitation

event scale in capturing rain-related soil moisture variability and timing, when compared with in situ data in the Sahel (Gruhier and Rosnay, 2008).

## 2.3 TMPA-v7 precipitation

The Tropical Rainfall Measuring Mission (TRMM) Multi-satellite Precipitation Analysis (TMPA) represents a partial-global coverage (50° S - 50° N) product of combined precipitation estimates. Three-hourly precipitation time-series of the TMPA

product (1998 - 2015, Huffman et al., 2007) at $0.25° \times 0.25°$ horizontal resolution are used to estimate locations of afternoon convective precipitation events in the study.

The TMPA algorithm (Huffman et al., 2007) involves the following steps: (1) - merging multiple independent passive microwave sensors, (2) - their inter-calibration to the TRMM Combined Instrument (TCI) precipitation estimates, (3) - further blending with preliminary calibrated infra-red products from geostationary satellites, and finally (4) scaling of the estimates to

match monthly accumulated Global Precipitation Climatology Center (GPCC) rain gauge data.

In this study we utilize the product version 7 (TRMM-3B42), which includes several modifications to the algorithm and additional satellite data (Huffman and Bolvin, 2014). Consistent with the soil moisture record, only 10 years (2002 - 2011) of JJAS precipitation data is used. To ensure similar solar forcing on the surface and boundary layer, the 3-h precipitation time-series for the present application are adjusted to LST (based on longitude) by taking the closest 3-h UTC time step. It is





important to note that any 3-h TMPA value is not referred directly to its nominal hour, but represents the average of the "best" overpass data within a 3 hourly window, centered around the nominal hour, i.e. +/- 90 min range. Variable time of the TMPA "best" data average is not expected to significantly affect our SMPC results.

## 3  Methods

### 3.1  Description of statistical framework after T12

The SMPC in this study is referred to as the relationship between the afternoon convective rainfall and soil moisture conditions in the few preceding hours. Using the method of T12, we examine whether afternoon rainfall is more likely on days when soils are (i) - untypically (relative to a control sample) drier or wetter than their surrounding, and (ii) - untypically drier or wetter than their temporal mean. Subsequently, the higher than expected probability of convective rainfall events to occur over spatially drier or wetter soils is referred to as *spatial SMPC*, while the higher than expected likelihood of convective rainfall events to occur over temporally wetter or drier soils quantifies *temporal SMPC*. The following paragraph describes criteria which are used to define a convective precipitation event, and evaluate soil moisture statistics antecedent to every event. The framework algorithm implemented in this study largely follows the method of T12 and is summarized in Fig. 2.

### 3.1.1  Definition of convective rainfall event (Fig. 2C)

We define a convective event location, $L_{max}$, as the location where accumulated afternoon precipitation between 15:00 - 21:00 LST exceeds a threshold of 6 mm. Then, locations of afternoon accumulated precipitation minima, $L_{min}$, are identified within a $5 \times 5$ pixel box ($1.25° \times 1.25°$)[1] centered at $L_{max}$ (Fig. 2C). The choice of a later accumulation time than in T12 (i.e. 15:00 - 21:00 LST instead of 12:00 - 21:00 LST) ensures that the soil moisture measurement at 13:30 LST precedes precipitation without introducing additional filters. The twice larger afternoon accumulated rainfall threshold than in T12 yields qualitatively similar results, though leads to a slightly higher mean SMPC significance over the domain. According to additional sensitivity tests, the choice of higher threshold values in the method mostly influences the amount of significant grid boxes linked to a reduction in the event sample size, yet does not qualitatively affect the dominant preference of the afternoon rainfall over specific soil moisture conditions (Petrova, 2017).

The following set of assumptions is used to improve the accuracy of the convective event sample. If one of the conditions is not fulfilled, an event is excluded from further calculation:

(1) - accumulated precipitation in the preceding hours (06:00 - 15:00 LST) in the entire $1.25°$ box must be zero;

(2) - elevation height difference within the event box must not exceed 300 m. The latter is done to minimize the effect of orographic uplifting on the rainfall variability. The resulting distribution of the orography mask is shown in Fig. 3a.

(3) - number of identified $L_{min}$ locations within one box must be 3 or more (for averaging reasons), and a negative rainfall gradient between $L_{max}$ and its adjacent four pixels must be present. These conditions were not considered in T12 and G15

---

[1]Following T12, a box size of $1.25° \times 1.25°$ is selected as minimum possible size to resolve soil moisture variability around the center of the box, taking into account the 50 km footprint of the AMSR-E soil moisture.





methods, and reduce erroneous events identified within or at the edge of propagating squall lines or large organized convective systems.

(4) - if boxes overlap, the event with larger afternoon accumulated precipitation value is retained.

### 3.1.2 Soil moisture statistics in event locations (Fig. 2D - E)

Once events are identified, soil moisture anomaly $S'$ measured prior to the precipitation event (at 13:30 LST) at $L_{max}$, $\overline{L_{min}}$ or any combination of the two can be stored and analyzed. $\overline{L_{min}}$ represents an average value of $S'$ measured in every identified $L_{min}$ location within a $1.25°$ event box. $S'$ has its climatological mean subtracted, calculated as a departure from the $\pm 10$ day mean over 10 years. To exclude contribution of a rain event onto the anomaly values, the year of the event is excluded from the climatological mean calculation. In order to investigate whether it rains over spatially wetter or drier soils, we calculate the

pre-event soil moisture gradient between $L_{max}$ and $\overline{L_{min}}$ scaled per 100 m, i.e. $Y_e = \Delta(S'^{L_{max}}_e)$ with the dimension of m$^3$ m$^{-3}$ 100 m$^{-1}$, where $\Delta$ - stands for gradient (Fig. 2D). To estimate, whether it rains over temporally wetter or drier soils we store pre-event soil moisture anomaly at $L_{max}$ location, i.e. $Y_e = S'^{L_{max}}_e$.

For every two $Y_e$ parameters we define the control sample $Y_c$, represented by an array of corresponding $Y$ values measured in the same $L_{max}$ and $L_{min}$ event locations in the same calender month, but on the non-event years. The measure of coupling

is then quantified by the magnitude of a difference between mean statistics of the event and control samples, $\delta_e = \text{mean}(Y_e) - \text{mean}(Y_c)$, and the measure of $\delta_e$ significance (Fig. 2E). Significance is represented by a percentile, $P_e$, of the observed $\delta_e$ in a bootstrapped sample of $\delta$ values that is observed by chance. For that $Y_e$ and $Y_c$ are pooled together and re-sampled without replacement 5000 times.

### 3.1.3 Definition of temporal and spatial SMPC (Fig. 2F)

Parameters of $\delta_e$ and $P_e$ calculated for the soil moisture gradients $\Delta(S'^{L_{max}}_e)$ prior to the event quantify preference of rain to occur over soils drier ($\delta_e < 0$, $P_e \leq 10 \%$ ) or wetter ($\delta_e > 0$ , $P_e \geq 90 \%$) than its $1.25°$ environment, and are referred to as negative or positive *spatial SMPC* respectively. The same parameters estimated for the temporal soil moisture anomaly $S'^{L_{max}}_e$ instead specify expressed preference of rain to occur over soils drier or wetter than its temporal mean, i.e. negative or positive *temporal SMPC* accordingly (definition as in G15).

In this study, estimation of $\delta_e$ and its significance $P_e$ for the spatial and temporal coupling is realized over $5° \times 5°$, $2.5° \times 2.5°$ and $1° \times 1°$ boxes. Aggregation of event statistics at a higher resolution than used in the global studies of T12 and G15 results in a smaller event sample size per grid box, yet allows a reduction of the potential influence of meridional or zonal gradient in the parameter statistics, i.e. makes the spread in underlying surface and atmospheric moisture conditions across the box latitudes smaller (Section 4.2). The latter is valuable for the interpretation of obtained statistics in terms of land cover and

atmospheric state. Hence, most of the study focuses on the smallest $1°$ spatial grid.



## 3.2 Statistics of convective events

Application of the algorithm to the 10 years of JJAS AMSR-E soil moisture and TMPA precipitation time-series yields 10131 afternoon rainfall events. The distribution of identified events over the domain at 1° and 5° grid is shown in Figure 3b and 3c respectively. The signature of orography and large-scale dynamic effects on event occurrence becomes evident only at the higher event-aggregation scale, thus giving an advantage to the highest horizontal resolution. Figure 3b shows that most events occur between 10° N and 18° N, and the occurrence maxima are zonally aligned. Two maxima are found over the central Sahel, covering the area between 10° W - 15° E - aligned with the mean position of the AEJ core (Figure 1b). Another two maxima are evident at about 22° E and 30° E, and are likely formed as a combination of orography-induced propagating convective systems and the orography mask applied in this study. The obtained distribution of identified rain events at 1° grid resolution is consistent with the observed distribution of intense MCS over the region (Mathon and Laurent, 2001).

## 4 Results of spatial SMPC analysis

### 4.1 SMPC at 5° horizontal resolution. Consistency to previous studies

We start our assessment by investigating the spatial soil moisture - precipitation coupling relationship. In agreement with the global-scale studies of T12 and G15, we find a dominantly negative spatial SMPC in the Sahelian domain at the 5° scale, i.e. a strong preference for convective rainfall events to occur over spatially drier soils (Fig. 4a). The majority of the 5° boxes (72 %) have percentile values $P_e$ lower than 10 %, implying a significant negative difference in the mean magnitude of soil moisture gradients $\Delta(S_e'^{L_{max}})$ prior to the events relative to their typical (non-event) state. No significant positive difference between event and non-event conditions is found at the 5° scale (Table 1).

Figure 5 further compares the percentage of the domain area with significant negative coupling identified in our study, T12 and G15. The differences arise due to disparities in the data sets and methodologies. As in G15, the weakest negative coupling signal in the Sahelian domain is obtained with the PERSIANN (Precipitation Estimation from Remotely Sensed Information using Artificial Neural Networks) data set (Hsu et al., 1997). This is possibly linked to the lower consistency between the PERSIANN precipitation and soil moisture variability in time (T12). On average, all the experiments summarized in Fig. 5 agree that afternoon precipitation occurs more often than expected over spatially drier soils in 42 % of the studied boxes, against only 4 % with a preference over spatially wetter soils.

The variability of spatial SMPC patterns among different data set combinations has shown to be quite strong over the globe and was not analyzed further in G15. We find, however, that in the Sahelian domain, areas of significant negative spatial coupling are fairly consistent. One of the most robust negative spatial SMPC signals is found in the south-western part of the domain (Fig. 4a,b). Fourteen out of 18 data set combinations summarized in Fig. 5, including this study, locate the cluster of the lowest percentiles roughly between 5 - 15° N and 10° W - 10° E (Fig. 5, crosses). This region occupies a relatively vast and flat area, associated with a reduced orographic forcing on convection development compared to the East, and a regional minimum in cold cloud occurrence (Laing and Carbone, 2008). The effect of large-scale disturbances like AEWs and AEJ on



convection, on the contrary, is expected to be stronger in the western Sahel than further East. However, this does not exclude or even favour higher sensitivity of convection triggering to soil moisture heterogeneities (Gantner and Kalthoff, 2010; Adler et al., 2011). The identified negative spatial SMPC relationship in the region is consistent with the recent observational- (Taylor and Ellis, 2006; Taylor, 2010; Lothon et al., 2011) and model-based (e.g. Gantner and Kalthoff, 2010; Garcia-Carreras et al.,

2011; Birch et al., 2013; Taylor et al., 2013) studies in the Western Sahel.

Another cluster of the lowest percentiles and the largest differences in soil moisture state between event and non-event days $\delta_e$ is identified in the south-east of the domain (Fig. 4a,b). The proximity to the Ethiopian Highlands and the presence of extensive seasonally flooded regions in this area makes it generally difficult to isolate effect of surface state on convection. This possibly led to less coherence in the spatial SMPC estimates identified in our study, G15, and T12 analyses (not shown).

Unlike in the western Sahel, no accurate estimates of the SMPC exist in this eastern region.

### 4.2   Robustness of the negative SMPC at higher 2.5° and 1° horizontal resolution

In order to better identify the factors and potential physical mechanisms that affect the magnitude and variability of the SMPC we reduce the event-aggregation scale to the finer 2.5° and 1° horizontal grid[2]. In particular, aggregation of the convective rainfall events and corresponding soil moisture statistics over the smallest 1° grid boxes can reveal more details on the effects

of land surface conditions on the SMPC.

The percentile maps obtained for the finer scales of event-aggregation are presented in Figures 4c and 4e. Despite the reduction in the amount of significant $\delta_e$ values, largely due to the decreased number of events in every box, negative spatial SMPC relationships remain dominant at the finer scales, and exhibit a similar spatial pattern as at the 5° resolution. The featured regions of significant negative coupling now scale down to the territories of Burkina Faso, Benin, parts of Ivory Coast, Ghana

and Mali (7 - 15° N, 10° W - 7° E) in the West, as well as South Sudan (5 - 13° N, 24 - 34° E) in the East. In total, 42 % (21 %) of the boxes reveal significant negative difference $\delta_e$ for the 2.5° (1°) grid resolution, versus initial 72 % at the 5° scale (Table 1).

The overall distribution of the $\delta_e$ does not change at the finer scales (Fig. 4d,f). However, multiple pixels with a positive $\delta_e$ emerge. For example, a small region enclosed between the Cameroon mountains and Jos Plateu (7° N; 8° E, Fig. 4f)

now indicates a higher likelihood of rainfall to occur over spatially wetter soils. The relationship, though non-significant, is plausible. This area includes part of the Niger river valley and represents a prominent location of intense convection and a local maximum of the cold cloud occurrence, linked to the initiation of convection at the lee side of the high terrain (Laing and Carbone, 2008). In total, 14 % of the 1° boxes reveal a positive $\delta_e$ shift, against less than 0.1 % for coarser 2.5° and 5° grids.

### 4.3   Evidence for "wetland-breeze" mechanism in the SMPC statistics

In areas like the Cameroon Mountains where orography or floodplains have an effect on deep convection development, persistent wet and dry surface moisture patterns may pre-exist or develop, and therefore, lead to the occurrence of stronger than usual spatial soil moisture gradients. In the SMPC statistics, such gradients occur as extremes in a given distribution of soil

---

[2]The SMPC statistics is calculated if at least 8 events in a box are present.





moisture gradients $\Delta(S_e'^{L_{max}})$ within a $1° \times 1°$ grid box. Here, $\Delta(S_e'^{L_{max}})$ is considered to be an extreme if it lies outside the $(Q_{25}$-$1.5 \times IQR, Q_{75}$+$1.5 \times IQR)$ range, where $Q_{75}$ and $Q_{25}$ are the third and first quartiles respectively, and the interquartile range (IQR) is the difference between them. The distribution and magnitude of the extreme soil moisture gradients identified in the domain are shown in Figure 6b.

We find that 28 % of all valid $1° \times 1°$ boxes, and thus $\delta_e$ values, are affected by extremes. A large part of the extreme soil moisture gradients are located in the regions of significant negative coupling in the West and East (Fig. 6b). In these areas, extreme $\Delta(S_e'^{L_{max}})$ lead to an overestimation of the SMPC magnitude, and in some cases appear to predefine its significance. Removal of extremes leads to a decrease in the number of boxes with significant negative spatial coupling by 30 %. However, in most cases the sign of the coupling remains unchanged. In the grid boxes where extreme $\Delta(S_e'^{L_{max}})$ affect the sign of the

SMPC, the mean and median of $\Delta(S_e'^{L_{max}})$ distribution have opposite signs (Fig. 6b, black dots).

    Further analysis shows that extremes tend to cluster around major rivers and wetland areas in the East and West (Fig. 6a,b), which supports our hypothesis. Strong positive soil moisture gradients are found around the Senegal river close to the coast, and on the lee side of the Cameroon Mountains. Strong negative soil moisture gradients are more numerous and seen all along the western flow of the Niger river, downwind of the permanent wetlands of Ez Zeraf Game Reserve and irrigated lands of the

Gezira Scheme in Sudan. The scatter of the extremes in the East may be related to the recurrent floods of the White Nile river.

    The identified sensitivity of the afternoon rainfall to the strong negative soil moisture gradients around water bodies is in agreement with the results of the observational-based study of Taylor (2010). Analyzing 24 years of Meteosat brightness temperatures over the Niger Inland Delta, he found that convection was initiated more often over and to the east of the wetland in the morning hours. However, later in the day meso-scale convective systems tended to develop and propagate away from the

wet areas towards drier soils, suggesting formation of deep convection and afternoon precipitation over negative soil moisture gradients. Similarly, observed by Alter et al. (2015), enhancement of rain to the East of irrigated land at $14°$ N, $33°$ E and its suppression over the Gezira Scheme itself is consistent with the location of negative (positive) extreme soil moisture gradients to the West (East) of the irrigated region (Fig. 6b). All the above suggests the potential relevance of thermally-driven "wetland-breeze" circulations on convection triggering as well as moist convection intensification over the drier soils adjacent to the

flooded areas.

### 4.4   Effect of propagation of deep convective events on the SMPC statistics in eastern and western domains.

Another physical effect that may influence the SMPC relationship is related to the propagation and evolution of meso-scale convective systems (not accounted for by the current algorithm). Previous studies indicate that an opposite SMPC relationship might be expected at early versus late stages of MCS development (Taylor and Lebel, 1998; Taylor and Ellis, 2006; Taylor

et al., 2010; Alonge et al., 2007; Clark et al., 2003; Gantner and Kalthoff, 2010). In this respect, a distinct strength or even sign of the spatial SMPC measure may result from separation of the rainfall events into those formed by a weaker and smaller MCSs - mostly found in the early afternoon - or by long-lived and organized MCSs - dominant during late afternoon hours (Mathon and Laurent, 2001). Differences in SMPC response to MCS life cycle are also expected to exist between the two regions of significant negative coupling, in the East and West. To characterize these differences, we analyze precipitation diurnal cycles





averaged over event days in the East and West first (Fig. 7), and then estimate sensitivity of the spatial SMPC to varying rainfall accumulation times (Fig. 8).

The Hovmöller diagram of rainfall averaged over 1000 event days in the Western domain (black rectangular in Fig. 6b) shows that intensification of the moist convection in the region is generally concentrated around main orographical features

(Fig. 7a,c). The peak in precipitation occurs at similar times across the domain, and thereby does not reveal expressed signature of the system propagation. Most of the MCS are therefore expected to be shorter-lived and smaller, suggesting that their dissipation locations would be found close to their initiation (Mathon and Laurent, 2001).

In the East, on the contrary, the strong south-western propagation component of moist convection dominates the zonal progression of the most intense rainfall during diurnal cycle averaged over 754 event days (Fig. 7d). A large number of MCS

initiate at the lee side of the Ethiopian Highlands and propagate westward undergoing cycles of regeneration and growing into a mature and organized MCS (Laing and Carbone, 2008; Laing et al., 2012). The emergence of an absolute rain rate maximum downwind of the permanent wetlands of the Ez Zeraf Game Reserve during afternoon hours indicates a strong influence of the flooded areas on moist convection intensification in the region (Fig. 7d,f). Consistent with the results of Taylor (2010) obtained for the Niger Inland Delta, the presence of wetlands in the Eastern domain is expected to increase the number of organized and

long-lived propagating MCS in the late afternoon, originating from either locally triggered MCS, i.e. formed at the dry land-wetland boundary, or from re-intensified pre-existing westward propagating systems. The identified location of the maximum rain rate westward from the permanent wetlands at 30° E is consistent with the increase in cold cloud occurrence observed by Taylor (2010) downwind of the wetlands of the Niger Inland Delta. The latter supports the presence of similar mechanisms operating in the Ez Zeraf Game Reserve. We may therefore expect a greater sample of long-lived and organized propagating

MCS to be found in the late afternoon hours in the Eastern than in the Western domain. Accordingly, the response of the SMPC statistics to propagating MCS is expected to be stronger in the East compared to the West. Figure 8, which shows the change of the SMPC parameter between different rainfall accumulation time periods confirms this hypothesis. For this assessment additional area in the Northern Sahel is considered (gray rectangular in Fig. 6b), as representative of a region where large-scale atmospheric and surface conditions differ from those of the East and West domains.

From Figure 8 it is seen that the earlier rainfall accumulation time periods, i.e. 12:00 - 18:00 UTC in the East and 15:00 - 21:00 UTC in the West and North, result in the strongest negative $\delta_e$ difference, and hence spatial SMPC relationship in all three domains. No positive $\delta_e$ values are found for these time periods, and the fraction of negative soil moisture gradients preceding rainfall events are: 62 %, 57 % and 55 % for East, West and North accordingly. Later accumulation times lead to a decrease in the magnitude and significance of the coupling parameter $\delta_e$, and an increase in its spatial variability across the

domains. These changes are associated with an increase in amount of the positive soil moisture gradients in the regions.

Despite the similarities, differences in the SMPC response exist between the domains. In the East, the spatial SMPC shows the strongest sensitivity to the rainfall accumulation time and switches the sign to a positive one for the 18:00 - 24:00 UTC period. In accordance with Fig. 7d, the 18:00 - 24:00 UTC period reflects the afternoon progression of the mature MCS formed during early afternoon hours at the Ethiopian Highlands and around wetlands. The large and organized MCS are known to be

more efficient in developing over wetter soils, associated to a well expressed BL moisture anomaly and higher MSE and CAPE





(Taylor and Lebel, 1998; Taylor et al., 2010) and, at the same time, might get suppressed over drier surfaces (Clark et al., 2003). These observations are consistent with those identified here, i.e. increase in fraction of positive $\Delta(S_e'^{L_{max}})$ in all the domains towards late afternoon hours, and the strongest SMPC response in the East.

The Eastern domain also exhibits the strongest negative $\delta_e$ of the three domains, when the earliest time period (12:00 - 18:00 UTC) is considered. This time period includes the rain rate maximum formed in the vicinity of wetlands and hence, is likely in conjunction with the triggering of convection by the "wetland-breeze" mechanism. As a result, extreme negative soil moisture gradients observed during this time period dominate the statistics of the identified strong negative SMPC. Additional analysis reveals that the majority of large and negative soil moisture gradients in all domains are linked to the rainfall events that are identified during the first afternoon time step (i.e 12:00 UTC and 15:00 UTC for the East and West respectively), and are therefore linked to weaker MCSs at the early stage of their development (not shown). The smaller and less organized MCSs have shown to be more sensitive to the thermally-induced surface convergence zones and are likely to develop over spatially drier soils, adjacent to the strong gradients (e.g, Gantner and Kalthoff, 2010). This knowledge is consistent with the strongest negative $\delta_e$ difference identified here and hence with the SMPC relationship during early afternoon times in all three domains.

## 5 Results of temporal SMPC analysis

### 5.1 Co-variability of the spatial and temporal SMPC

The presence of a negative spatial SMPC with inherent features of the physical effects identified above support a potential role of thermally-induced circulations for moist convection intensification over spatially drier soils. Higher probability of "breeze-like" circulations to occur over the strong soil moisture gradients is expected when the soil moisture content in $L_{max}$ is relatively low. This condition would allow a stronger buoyancy flux in $L_{max}$ and at the same time a larger thermal contrast between $L_{max}$ and its surrounding. To explore on the latter hypothesis, we analyze soil moisture conditions prior to a rain event in $L_{max}$. By analogy to the spatial SMPC, we estimate the soil moisture anomaly $S_e'^{L_{max}}$ prior to the event and its difference $\delta_e$ to the typical state, i.e. temporal SMPC.

Analysis of $S_e'^{L_{max}}$ and its $\delta_e$ indicates a strong preference for rainfall events to occur over soils that are drier than their temporal mean (Fig. 9a) and drier than usual (Fig. 9b). The percentile values $P_e$ lower than 10% are found in 67 % of the studied $1°$ boxes (Table 1). The latter implies that a temporally negative SMPC dominates over the domain, which reaffirms the co-existence of the negative spatial and temporal coupling identified by G15, but at a finer $1 \times 1°$ resolution.

The question remains whether the two coupling relationships are independent of one another. To answer this question we calculate the Spearman rank correlation coefficient[3] event-wise between the soil moisture anomaly $S_e'^{L_{max}}$ and soil moisture gradients $\Delta(S_e'^{L_{max}})$ in every $1 \times 1°$ box. The correlation map in Figure 9c shows that a high and significant correlation exists between $S_e'^{L_{max}}$ and $\Delta(S_e'^{L_{max}})$ anywhere in the domain. The mean correlation of 0.47 over the domain supports the existence of relatively strong and positive monotonic relationship between the magnitude of spatial soil moisture gradient and

---

[3]Spearman correlation is a measure of monotonic relationship. Therefore, zero or low correlation value does not imply zero relationship between two variables.





soil moisture anomaly measured in $L_{max}$. For comparison, the mean correlation estimated between soil moisture gradients and mean soil moisture anomaly over the $1.25°$ event box is small (0.13). All the above suggests that in the North African region the spatial and temporal SMPC relationships, as defined by the current framework, are not independent of each other.

The strong and positive correlation ($in\ time$) identified between the soil moisture anomalies and gradients also yields a regional co-variability of the SMPC patterns. The *spatial* correlation between the two coupling distributions is high (0.64). The largest magnitudes of both $S_e'^{L_{max}}$ and $\Delta(S_e'^{L_{max}})$ parameters and their corresponding $\delta_e$ measures are found in the southern part of the domain. These regions are generally characterized as the areas of higher BL moisture and rainfall frequency, and therefore higher variability of soil moisture in time and space.

Mechanistically, the presence of the temporally negative SMPC in the areas of the highest BL moisture in the domain (lifting condensation level (LCL) is shown, Fig. 10a), is consistent with a higher relevance of mechanisms associated with the BL growth for convection initialization in regions of higher CAPE and lower convective inhibition (CIN) (Klüpfel et al., 2011; Gantner and Kalthoff, 2010; Adler et al., 2011). In this way, larger deviations of the soil moisture amount from its climatological mean and typical value, i.e. $\delta_e$, would imply presence of a stronger than usual thermals, which can easier overcome CIN and release CAPE (Klüpfel et al., 2011). In combination with a strong negative spatial gradients, these strong thermals can initiate breeze-like circulations, creating more favourable conditions for bringing BL up to the LFC, especially over the southern regions, where BL moisture is in abundance. A slight increase of the LCL in the South, associated with a decrease of BL relative and specific humidity (not shown) on event days compared to the typical state is seen in Fig. 10b (red shading). This observation supports the relevance of drier surface conditions for convection intensification as opposed to variations in BL water vapour amount prior to the events.

A different picture is observed over the drier latitudes of Northern Sahel at the Sahara margin. At these latitudes the northward excursion of moist monsoon air is shown to favour convective activity (Barthe et al., 2010; Cuesta et al., 2010). The estimated difference in LCL prior to the events relative to the typical state indicates that a significantly lower than usual LCL, associated to a significantly higher amount of water vapour in the BL (not shown), is present on the event days over the dry regions (Fig. 10b, blue shading). This result is consistent with the previously reported decisive role of low-level moisture on MCS evolution in the drier Sahelian regions (Klüpfel et al., 2012).

Considering also the relatively large number of dry days (10 days on average) preceding rain events in the North, it is less likely that underlying surface heterogeneity caused by a previous rainfall could have an influence on convection development on the event day. In the case study of Klüpfel et al. (2012) MCS was initiated due to the arrival of the cold pool and convergence zone emanated by a remote convective system hundreds kilometers away. Similar mechanisms may play a role in moist convection development in Northern Sahel.

## 5.2 Role of rainfall persistence

In the context of this study, the drying of the soil prior to the rainfall events might be considered as the primary process that underlies the magnitude of both SMPC relationships, and helps to explain the opposite sign of the temporal coupling identified in the Sahelian region as compared to the temperate latitudes and wet climates (G15).





Consistently to the observed 2 to 4 day periodicity of rainfall in Western Africa (Laing et al., 2012; Taylor and Lebel, 1998), 2 to 3 dry days (rain <1 mm) on average are found to precede each convective event day over southern latitudes, suggesting a strong drying of the upper soil layer in the event locations prior to the rain. The number of dry days reaches 10 over the dry and deserted regions in the North. Following the analysis of Schwendike et al. (2010) an almost complete recovery of the pre-rainfall surface moisture conditions may be expected in 2-3 days following the rainfall. Schematically, this typical variability of rainfall and soil moisture might be illustrated as a sequence of daily rain events separated by the periods of drying (Fig. 11a). From the Figure it is seen that prior to the rain events the soil dries out, and soil moisture reaches certain minimum value $S_{min}$. The climatology value $S_{clim}$ of soil moisture in the same location, however, is expected to be higher than any $S_{min}$ in most of the cases, as it includes all, dry and wet event days. Hence, when subtracted from the climatological value, a soil moisture measured prior to the event will very likely yield a negative anomaly - $S_e'^{L_{max}}$, especially when averaged over many events. Therefore, a negative correlation between soil moisture anomaly and rainfall might be expected. Though discussed in the framework of North Africa, similar behaviour might be expected in other water-limited regions of the world.

A different situation might occur in the wet temperate latitudes, where the variability of rainfall is to a large extent linked to fluctuations between passage of a cyclone and a blocking situation (Schär et al., 1999). Such a behavior might be illustrated as a multi-day sequence of rain events, associated with precipitation persistence as defined by the persistence in the weather regimes (Fig. 11b, see also Fig. 2 in Hohenegger et al. (2009)). During these periods soil moisture increases and remains relatively high. Hence, a higher fraction of events might be expected to occur over soils that are wetter than usual, resulting in a positive soil moisture anomaly $S_e'^{L_{max}}$ prior to the event. The above relationship is consistent with the negative spatial but positive temporal SMPC, identified in G15.

The modulation of the SMPC sign depending on the large-scale weather regime was studied e.g. by Boé (2012) over France. The analysis showed that the synoptic blocking situations generally associated with drier conditions lead to a negative SMPC, while positive correlation of rainfall to drier soil conditions was observed in wet weather regime. Similarly, most pronounced effect of negative soil moisture gradients on convection initiation over Europe and a higher correlation of the gradients to land surface temperatures was observed for the period with less antecedent rainfall (Taylor, 2015).

## 6 Summary and conclusions

In this study, the soil moisture - precipitation coupling (SMPC) relationship in the northern African region is investigated at $1°$ horizontal resolution using the probability-based approach of T12 and 10 years of satellite-based soil moisture and precipitation data. Specifically, we distinguish and analyze the temporal and spatial effects of soil moisture on afternoon convective rain.

We find that in the North African region spatial and temporal effects of soil moisture on afternoon precipitation are negative and are not independent of one another. The negative sign of the temporal coupling in the semi-arid conditions of the Sahelian environment is not unexpected. The drying of the soil for several days prior to the rainfall events is likely to underlie the preference of rain to occur over temporally drier soils, and additionally may play a role in the opposite sign of the temporal coupling as compared to the positive relationship identified in wetter climates by G15. For the same reason, the predictability



potential of the temporal effect on rainfall in the North African region is expected to be lower than in wet climates, while the spatial effect on the contrary is likely to have more relevance for predictability in the semi-arid regions.

The co-existence and co-variability of negative temporal and spatial SMPC across the Sahel supports the relevance of meso-scale spatial variability in soil moisture for moist convection development. Furthermore, it also hints on the relevance of processes associated with the dominance of sensible heat flux and boundary layer growth on convection initiation. In particular, the identified preference of rainfall to occur over temporally drier soils and strong negative soil moisture gradients might be considered as the most effective combination to maximize both the buoyancy and moisture flux at the event location through formation of the thermally-induced circulations, and hence lead to a higher probability of convection development. Schematic representation of the moist convection intensification by the breeze-like circulations formed under the co-existence of the two SMPC effects is illustrated in Figure 12.

The analysis of the BL moisture conditions (here, LCL) preceding the rainfall events suggests that the co-existence of two coupling effects, and hence potential role of "breeze-like" circulations on convection development is expected to be more relevant in the South of the domain, where BL moisture is in abundance. In the drier northern latitudes the variability of BL moisture, associated to intrusions of moisture from the south, seems to be more decisive.

Analysis of the spatial SMPC measure as well as factors which can affect its magnitude and variability in particular reveals two "hot spots" of significant negative spatial coupling: in the Western African domain (7 - 15° N, 10° W - 7° E), and South Sudan in the East (5 - 13° N, 24 - 34° E). In the Western domain, the negative spatial SMPC signal is indicated to be more robust. In the East, the spatial coupling is found to be largely modulated by the presence of wetlands and is susceptible to the amount of longer-lived propagating MCS. The number of propagating and mature MCS in the East increases towards late afternoon. Accordingly, changing the rainfall accumulation time period from early to late afternoon leads to a loss of significance of the spatial SMPC and a switch of its sign from the negative to the slightly positive one. Conversely, in the West, the majority of convective systems might be expected to be shorter-lived, and therefore smaller and less organized. In this region, negative spatial SMPC varies less with the selected afternoon time range.

Another factor which affects the magnitude and distribution of the spatial SMPC is related to the presence of extreme soil moisture gradients formed in the vicinity of wetlands and irrigated land. We find that removal of extremes leads to a decrease of the number of boxes with significant negative spatial coupling by 30 %. Concurrently, the identified sensitivity of the afternoon rainfall to the strong gradients in soil moisture adjacent to wetland areas hints on the relevance of wetland-breeze mechanism on convection intensification over spatially drier soils.

Following our analysis, a number of potential improvements to the framework might be summarized. Apparent non-local effects of water bodies and strong elevation height differences, that are originally excluded by the method, hints on the potential gaps in the filtering procedure and emphasizes potential role of moist convection evolution and propagation that are neglected by the method. The presence of wetland regions itself, as we have shown, complicates interpretation of the SMPC relationships. The uncertainty estimates of the soil moisture parameter derived over the recursively flooded regions are still missing.

Notwithstanding these limitations, this study demonstrates that the observed SMPC statistics is consistent with a number of physical effects and agrees on the sign of the SMPC suggested by previous case- and modelling studies. The identified





link to the wetland areas and rivers is only evident at the highest considered $1°$ event-aggregation scale, hence indicating the advantage of the finer scale over the coarser $5°$ grid. The SMPC "hot spots" identified in the present study may represent the regions where predictability skill of soil moisture on moist convection might be higher. The knowledge on the regional variability of the SMPC presented here can be further used in drought and climate change research, observational campaigns and GCMs validation.

*Competing interests.* The authors declare that they have no conflict of interest.

*Acknowledgements.* The authors would like to thank Max Planck Institute for Meteorology (MPI-M) and International Max Planck Reasearch School (IMPRS) for providing facilities, material and scientific support which made publication of this paper possible. The authors would like to acknowledge Christopher Taylor, Benoi Guillod and Alexander Mahura for helpful comments on the study, and Stephan Kern for data support. We also thank George Huffman and Robert Parinussa for their clarifications related to TMPA and AMSR-E data products respectively.





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





**Table 1.** General statistics of the average event number, percentile $P_e$ and $\delta_e$ difference estimated at various scales for three soil moisture parameters: soil moisture gradient $\Delta(S_e'^{Lmax})$, temporal soil moisture anomaly $S_e'^{Lmax}$, and (not presented in the methodology) soil moisture variance over the 1.25° box, $\sigma S_e^{1.25}$. Percentiles $P_e < 10$ % ($> 90$ %) indicate significant negative (positive) $\delta_e$ difference, and hence negative (positive) SMPC relationship.

| $Parameter$ | $Scale$ | $\overline{Num_e}$ | $P < 10$, [%] | $P > 90$, [%] | $\delta_{ev} < 0$, [%] | $\delta_{ev} > 0$, [%] |
|---|---|---|---|---|---|---|
| $\Delta(S_e'^{Lmax})$ : | $5 \times 5°$ | 309 | 72 | 0 | 92 | <0.1 |
| | $2.5 \times 2.5°$ | 84 | 42 | <0.1 | 73 | 0.1 |
| | $1 \times 1°$ | 17 | 21 | <0.1 | 43 | 14 |
| $S_e'^{Lmax}$ : | $1 \times 1°$ | - | 67 | 0.8 | 92 | 8 |
| $\sigma S_e^{1.25}$ : | $1 \times 1°$ | - | 33 | 3 | 78 | 22 |



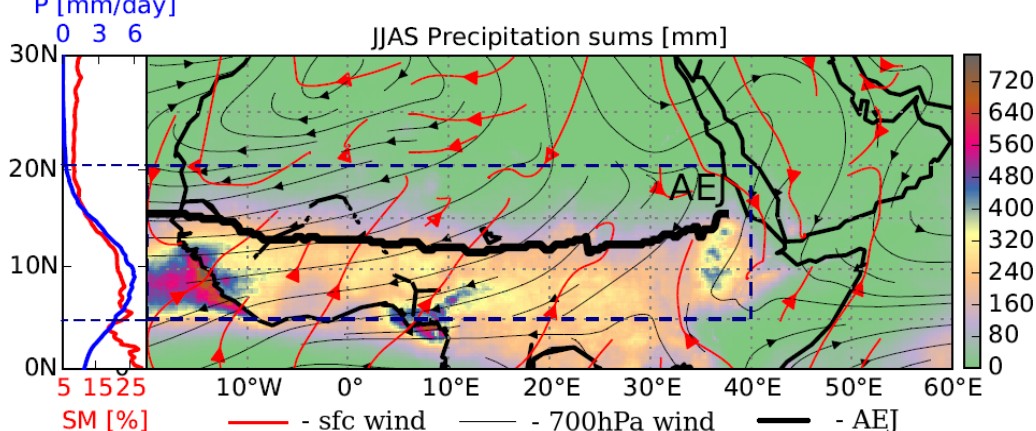

**Figure 1.** JJAS TMPA precipitation (shading), mean surface (red streamline) and 700 hPa (black streamline) ERA-Interim wind climatology averaged over 2002 - 2011 period. The black thick line shows mean location of the African Easterly Jet (AEJ). Inset plot to the left indicates zonal means of daily AMSR-E soil moisture (red) and TMPA precipitation (blue) climatology. The dashed rectangular shows the boundaries of the study domain. Wind data from the global atmospheric reanalysis ECMWF product ERA-Interim (1979 - present, Dee et al. (2011)) is re-gridded from the original T255 ($\sim 0.7°$) to $0.25°$ spatial resolution.





**A**  INPUT

- AMSR-E SM 13:30 LST
- TMPA precip. 3-hr UTC
- ETOPO orography

**B**  DATA FILTERING

- Orography < 300m / 1.25°
- Optical depth < 0.8
- Water < 5% / 0.25°

- Orography < 300m / 1.25°
- UTC → LST based on longitude (no interpolation)

**C**  EVENT DEFINITION  ASSUMPTIONS

Lmin

Lmax

Pr2

5 x 5 pixel
(1.25 x 1.25°)

- $Pr1 = \sum 06 - 15\ LST = 0\ mm$
- $Pr2 = \sum 15 - 21\ LST > 6\ mm$
- SM at 13:30 LST precedes rain

NOTATIONS

- $Lmax_i\ (t_i, lat_i, lon_i) = max(Pr2)$
- $Lmin_i^k(t'_i, lat'_i, lon'_i) = min(Pr2)$

**D**  OUTPUT PARAMETERS

- Soil moisture gradient, $\Delta(S_e'^{Lmax}) = S_e'^{Lmax} - \overline{S_e'^{Lmin}}\ /\ 100m$
- Soil moisture anomaly, $S_e'^{Lmax}$

**E**  BOOTSTRAPPING

$\overline{Y_e}$  $\overline{Y_c}$

$\partial_e$

$\frac{\partial (\overline{Y_{shuffled}})}{\partial_e}$

SM parameter     percentile

10  $P_e$  90

**F**  COUPLING MEASURES

- How different from control?  $\delta_e < 0$ — rain over drier soils
  $\delta_e > 0$ — rain over wetter soils
- How significant is difference?  $P_e < 10\%$ — negative coupling
  $P_e > 90\%$ — positive coupling

**Figure 2.** Schematic of data post-processing and statistical framework protocol implemented in the study.





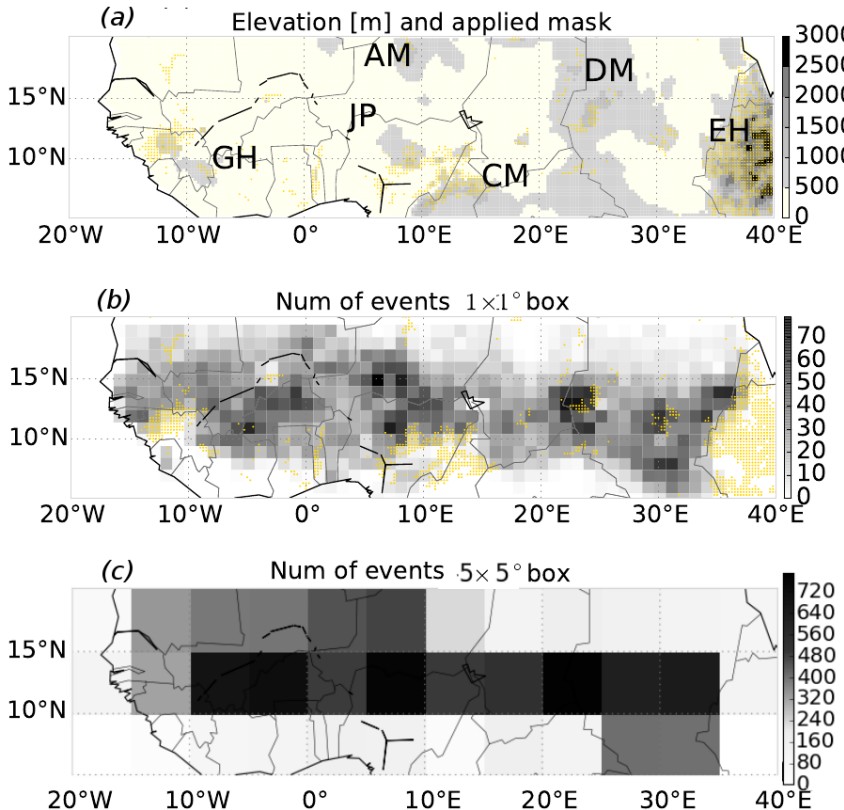

**Figure 3.** (a) - Elevation map based on 1 Arc-Minute Global Relief Model data ETOPO1 (Amante and Eakins, 2009) (grey shading) and orography mask used in the study (golden shading). Main orographic features of the region are: AM - Air Mountains, DM - Darfur Mountains, EH - Ethiopian Highlands, CM - Cameroon Mountains, JF - Jos Plateau, GH - Guinea Highlands. (b-c) - Number of events in every (b) $1° \times 1°$ and (c) $5° \times 5°$ box (gray shading) and applied orography mask at the $0.25°$ horizontal scale (golden shading).





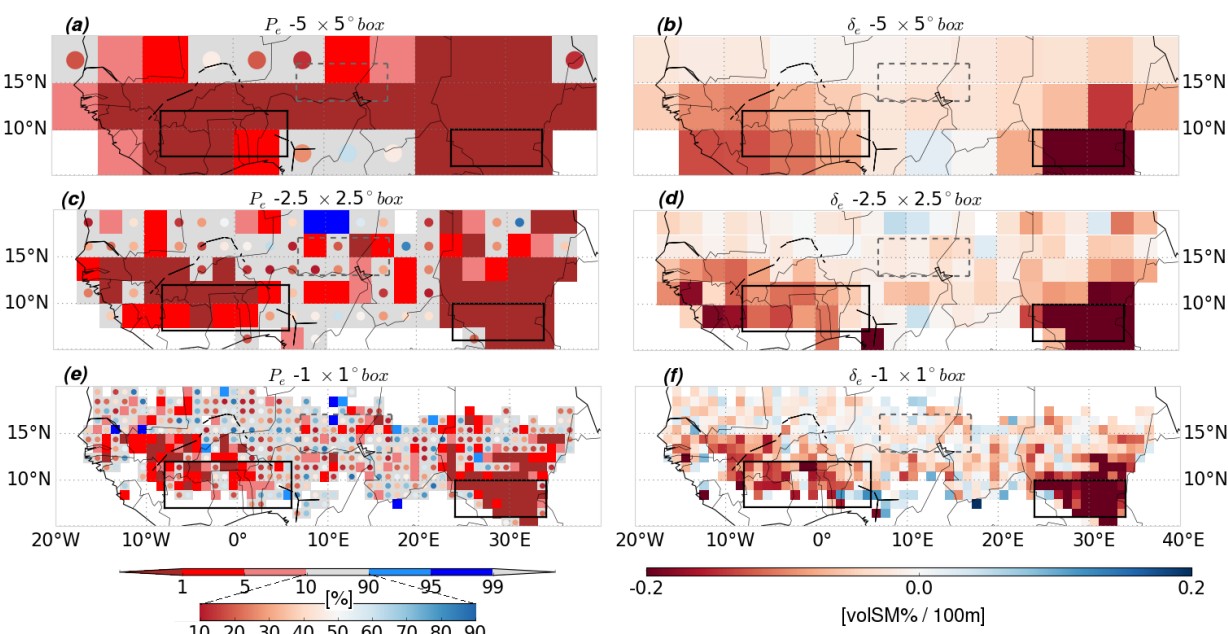

**Figure 4.** Distribution of percentiles $P_e$ (*left*) of the observed $\delta_e$ difference (*right*), estimated over $5 \times 5°$, $2.5 \times 2.5°$ and $1 \times 1°$ boxes. Percentiles $<10\%$ indicate significant negative coupling, i.e rain over spatially drier soils, and percentiles $>90\%$ - significant positive coupling, i.e. rain over spatially wetter soils. The percentile values lying outside the significance range (10 - 90 %) are illustrated by circles. Black and grey rectangulars on the maps indicate featured domains selected for an in-depth analysis.

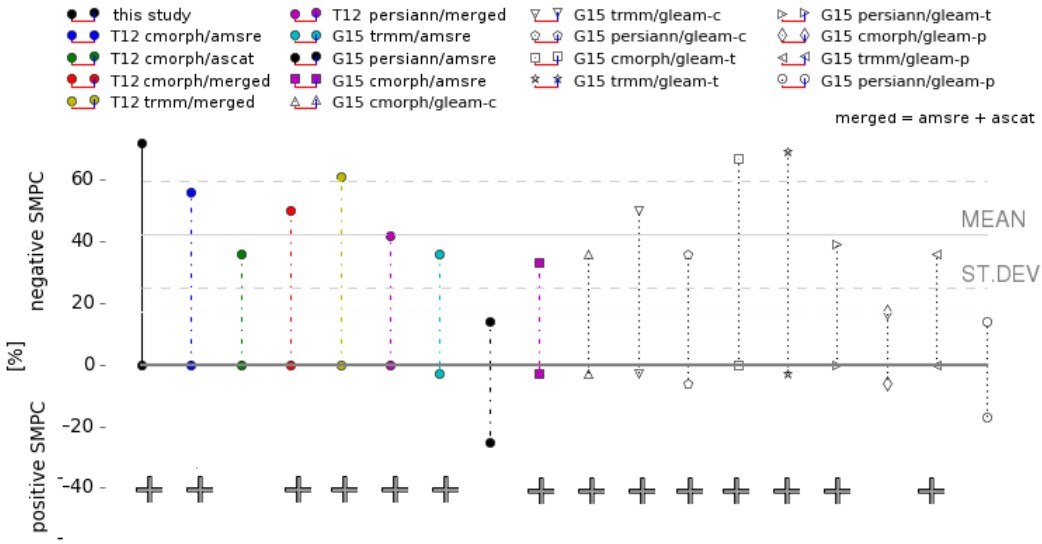

**Figure 5.** Percentage of $5° \times 5°$ grid boxes with significantly negative ($P_e < 10$ %) or positive ($P_e > 90$ %) spatial SMPC over Sahelian domain in this study and previous studies of T12 and G15. Various markers and colors represent different data set combinations used in T12 and G15. Colourless markers indicate soil moisture derived from GLEAM model with precipitation input from TRMM, CMORPH or PERSIANN datasets, referred as GLEAM-T, GLEAM-C and GLEAM-P respectively. Mean and st.dev. are calculated for the negative SMPC only. Following visual inspection, the experiments, in which significant negative SMPC relationship exists in the western region of the Sahelian domain are indicated by + markers.




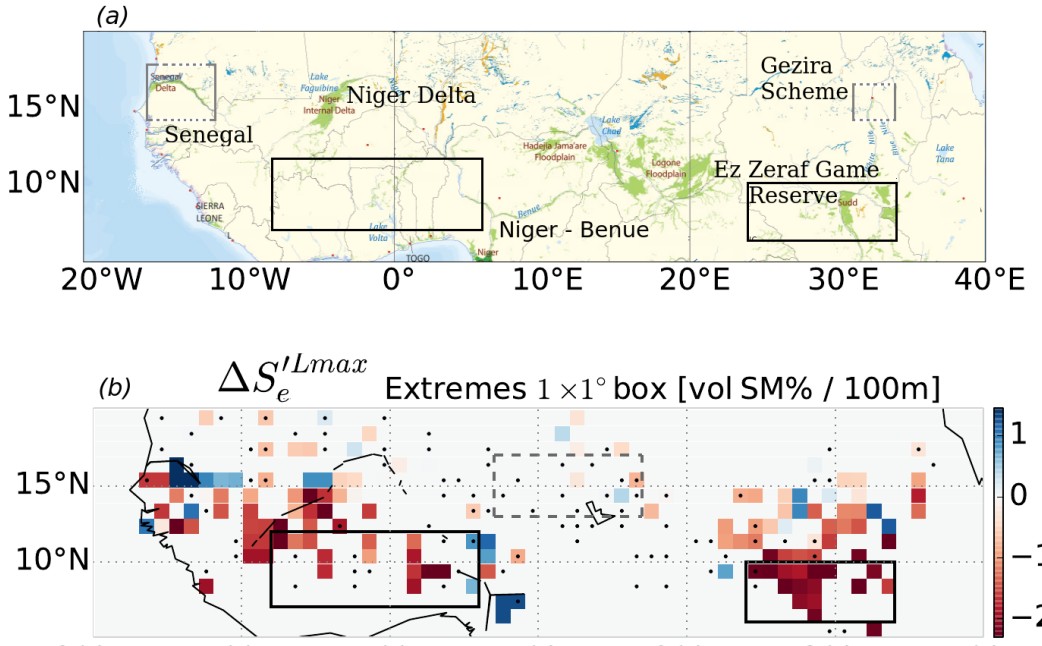

**Figure 6.** (a) - Major river flows (blue) and river flood planes (green) of the northern African domain [Adopted from Fig.1.4 of the Africa Water Atlas (UNEP, 2010). (b) - Distribution of soil moisture gradient $\Delta(S_e'^{Lmax})$ extremes in the corresponding event sample of a $1° \times 1°$ box. $\Delta(S_e'^{Lmax})$ is considered to be an extreme if it lies outside the $(Q_{25}$-1.5 $\times IQR, Q_{75}$+1.5 $\times IQR)$ range, where $Q_{75}$ and $Q_{25}$ are the third and first quartiles respectively, and the interquartile range (IQR) is the difference between them. Black dots indicate boxes, in which $\Delta(S_e'^{Lmax})$ sample mean and median have opposite signs.





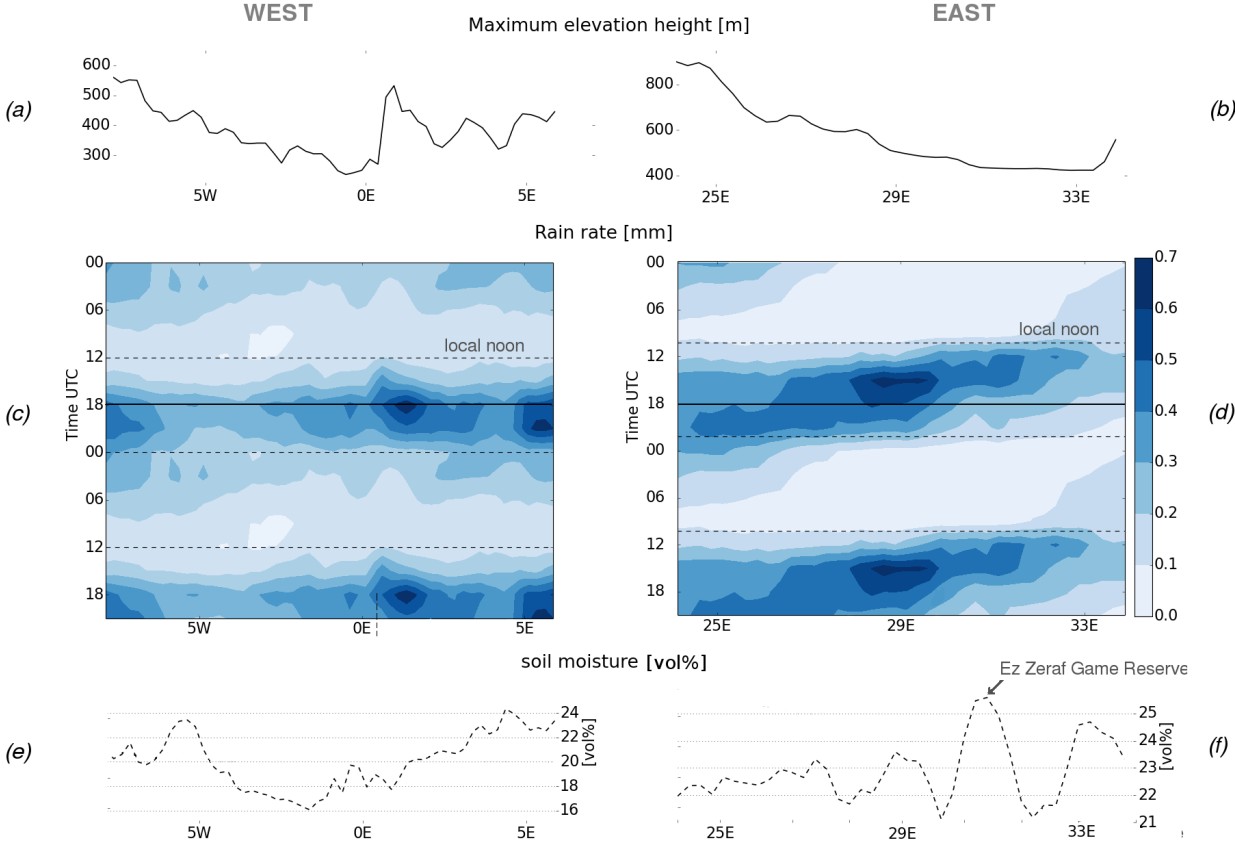

**Figure 7.** (a),(b) - Longitudinal cross-sections of maximum elevation height in the Western and Eastern domains respectively, (c),(d) - diurnal cycles of the rain rate averaged over event days and domain latitudes, and (e),(f) - Longitudinal cross section of soil moisture averaged over domain latitudes. Location of the Ez Zeraf Game Reserve permanent wetlands is marked by an arrow. All the times are given in UTC. Note, the UTC+2 hour difference to LST in the East.




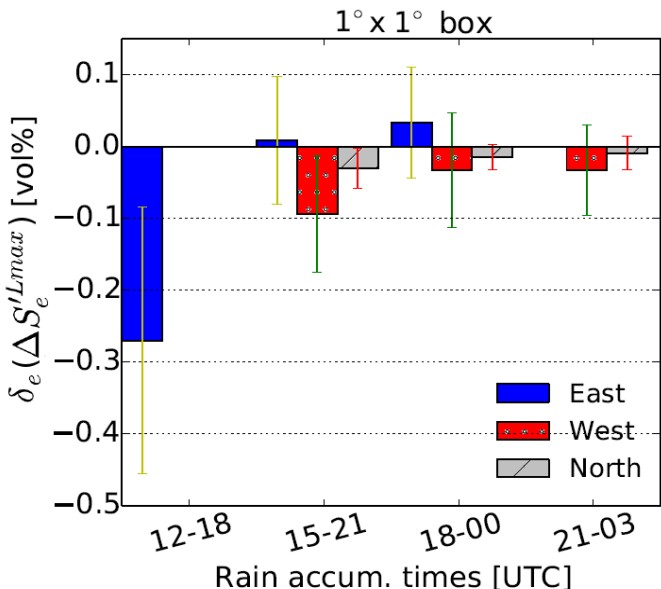

**Figure 8.** Value of the coupling measure $\delta_e$ calculated for various afternoon rainfall accumulation times, and averaged over selected domains, i.e East (6 - 10° N, 24 - 34° E), West (7 - 12° N, 8° W - 6° E) and North (14 - 17° N, 7 - 14° E). Locations of the domains are shown in Fig. 6b. Error bars indicate one std.dev. of $\delta_e$ values in every domain. Note, that all times are indicated in UTC.





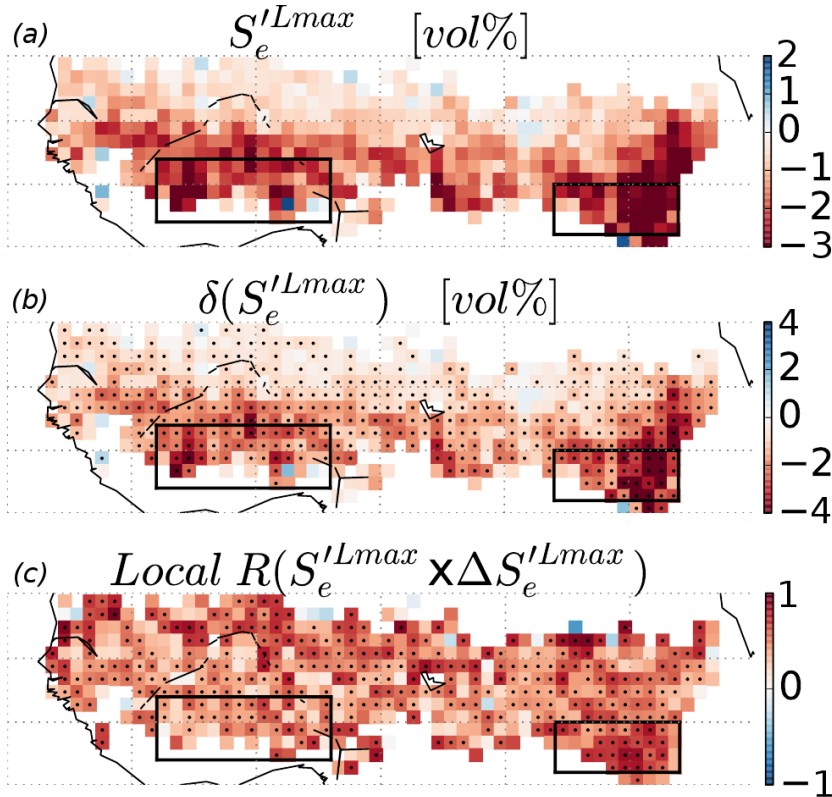

**Figure 9.** Distribution of the (a) temporal soil moisture anomaly $S_e'^{Lmax}$ in event locations and (b) its difference to the typical non-event conditions, $\delta_e$, averaged over $1° \times 1°$ boxes; and (c) *Local* Spearman rank correlation coefficient calculated event-wise between soil moisture anomaly $S_e'^{Lmax}$ and spatial soil moisture gradients $\Delta S_e'^{Lmax}$ in every $1°$ box. Significant $\delta_e$ values with percentiles $P_e$ below 10 % (above 90 %) and correlation coefficients with $p$-values lower than 0.05 are indicated by black dots.



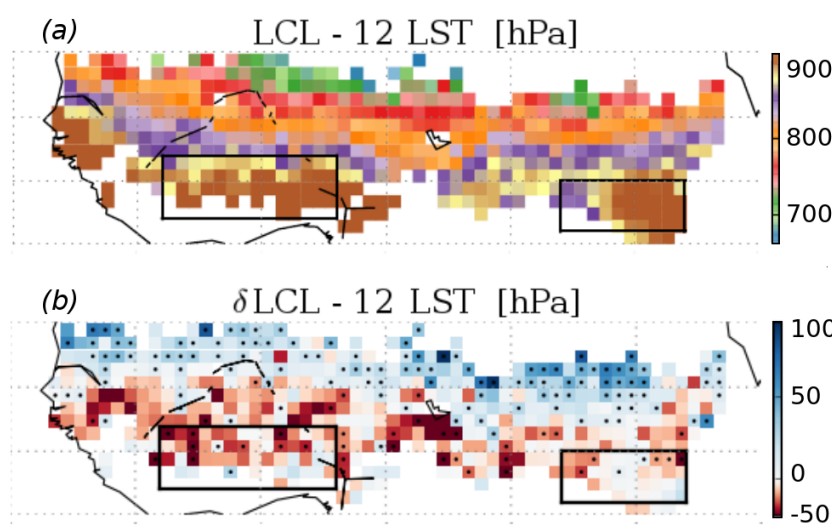

**Figure 10.** (a) - Lifting condensation level (LCL) value derived from the 6-hourly ERA-Interim temperature and specific humidity profile and surface pressure data on event days at 12:00 LST and averaged over the $1° \times 1°$ box. (b) - Corresponding $\delta_e$ difference of the mean LCL prior to the events relative to their typical state. The dot indicates significant $\delta_e$ values with percentiles $P_e$ below 10 % (above 90 %). The positive (negative) $\delta_e$ values indicate lower (higher) than usual LCL.



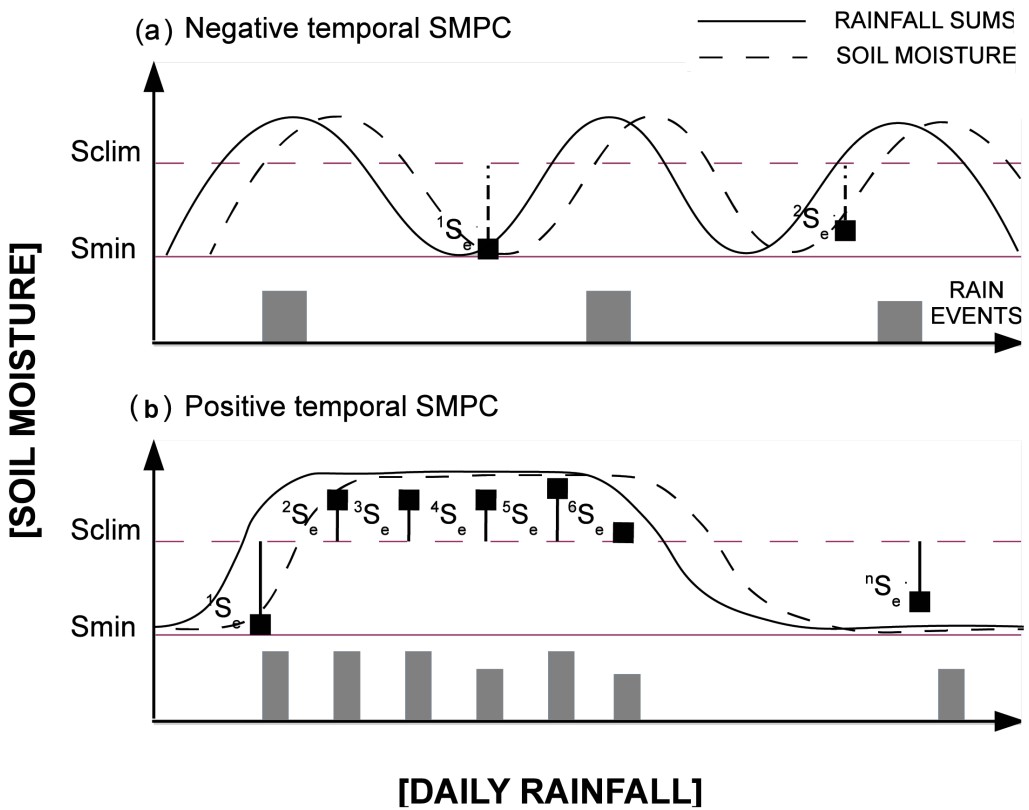

**Figure 11.** Conceptual diagrams of the relationship between daily rainfall occurrence and associated to it surface moisture variability in time as representative for (a) West Africa and temporally negative SMPC, and (b) Central and Northern Europe and positive temporal SMPC.





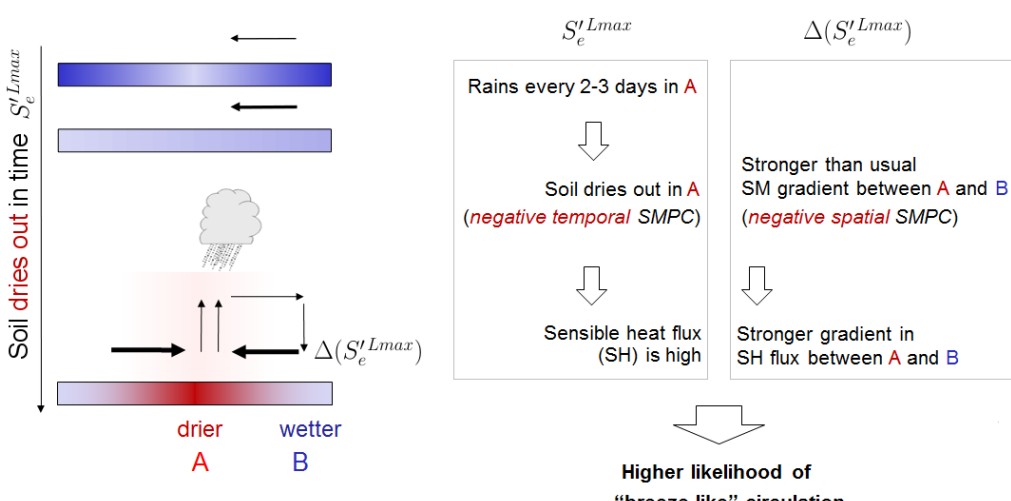

**Figure 12.** Conceptual diagram, illustrating intensification of moist convection by the initiated "breeze-like" circulations under favourable conditions of co-existing negative spatial and negative temporal SMPC effects.