# Peer review of "Regional co-variability of spatial and temporal soil moisture precipitation coupling in North Africa: an observational perspective."

_Hydrology and Earth System Sciences, 2017_

## Referee Comment (RC1) · Anonymous Referee #1 · 26 Oct 2017

The paper uses observations to explore the impacts of both spatial and temporal soil moisture variability on rainfall in North Africa. The authors use existing methodologies, but by applying them at much higher resolution they are able to explore feedback mechanisms, and their regional variations, in considerably more detail than in past global assessments. The authors find negative spatial and temporal soil moisture – precipitation feedbacks across the region, which are correlated with each other (implying co-dependence). They focus on two particular hotspots of this feedback, and discuss the role of wetlands, the size and propagation of typical convective systems in each region, and the role of rainfall persistence on their results.

[Figure]

Overall the paper is novel, timely and interesting, and the approach could (and should) be applied to other regions in the future as well. Unfortunately the paper is not always presented as clearly as it could be, partly in the structure, and partly the language. This means that I am left a bit unclear about some of the conclusions the authors reach, how they fit together and whether they are all well justified. I would therefore recommend the paper for publication once the following major revisions have been addressed.

Major comments

Presentation:

I found the paper a bit disjointed to read. I think there are lots of interesting ideas, but it jumps a bit from one thing to the other. I wonder if the results could be reorganised to make it easier to read. One suggestion is to first describe the feedbacks, i.e. show both the spatial and temporal soil moisture – precipitation coupling (figures 4, 5, 9, maybe 10), and then have a section talking about processes. The description of processes should then be consistent with both the spatial and temporal feedbacks you observe. The language throughout could be improved a bit as well (I have included some suggestions below, but my comments are not comprehensive). Finally, some figures I think could also be improved (my suggested changes are in the minor comments).

Wetlands (S 4.3):

Of the results, this is the part that I found most confusing, and therefore least convincing. Some points:

- Your definition of extreme is very confusing, it would be easier if it were expressed simply as the values above/below given percentiles (which is more or less what is done here, as far as I can tell, but with a fixed offset as well).

- I agree that features such as wetlands are likely to lead to extreme soil moisture gradients, but you might also expect the temporal variability to be low (at least over the wet part). Given you calculate temporal statistics, why not use this as an additional

criterion? i.e. find locations with high spatial and low temporal variability.

- Following on from this, in Fig 6b the dots (where the median and mean are opposite signs) do not really match the extremes (colours). I'm therefore not sure if the statement is justified that "extreme [soil moisture gradients] . . . in some cases appear to predefine its significance".

- The map (fig 6a) is not all that useful, as it is hard to see the exact locations and extent of topographic and wetland regions. It is therefore hard to tell if the conclusions in the last two paragraphs are really substantiated. For example, the region of extremes in the south East is very large, and I can't really tell how much of it falls on actual wetland. Similarly, the Gezira scheme in the map is quite small, and it's quite hard to tell what points on the significance contours you are linking to it.

- More mechanistically, could the wetlands not be partly the cause of the covariability between the spatial and temporal feedbacks? The wetlands are approximately spatially and temporally constant, therefore when the soils surrounding it are drier than normal, this will always represent both a temporal and spatial negative soil moisture anomaly.

Reasons for covariability of spatial and temporal SMPC:

The discussion here is again a bit muddled. In the conclusions you state "the drying of the soil for several days . . . may play a role in the opposite sign of the temporal coupling as compared to the positive relationship in wetter climates". It is worth noting that in G15 the temporal coupling is positive across most of the globe, including in other arid and semiarid regions (northern India, Australia, Saudi Arabia, etc.). The 3-4 day variability of rainfall in West Africa driven by African easterly waves is a factor, as is pointed out. I think, however, that the primary factors is probably that this is a high CIN/high CAPE environment, therefore anything that helps overcome the CIN will enhance rainfall. This is mentioned in the text when comparing the southern and northern part of the domain, but I suspect it can also explain the differences with other areas of the globe.

Minor comments

P2, L8: 'have a direct effect on the resulting sign'. It would be useful to explicitly state what the sign they find is (i.e. positive temporal, negative spatial)

P4, L9-10: 'on the northern flank. . .' In the diagram it looks like the gradients go across the ITCZ, as opposed to just being on the northern side?

P6, L29-30. You say you need at least three values of Lmin – do you take the three lowest? (presumably, unless Lmin is 0, there is only one minimum value). I am also confused by the criterion that 'negative rainfall gradient between Lmax and its adjacent four pixels must be present'. If Lmax is the maximum, then the gradient with the 4 pixels surrounding it will always be negative – I suspect I am misunderstanding this line.

P8, L9: '. . .orography mask applied in this study'. Do you mean the maxima are produced by the fact that you are masking the region around the maxima? This sentence is not very clear.

P8, L31: I wonder if the different datasets agree more also because precipitation retrievals themselves are more consistent over flat terrain, while they are likely to disagree more over complex topography.

P9, L19: might be worth mentioning that the significant correlations lie exactly on the semiarid transition zone between forest (in the south) and the desert. Also, have you considered the impact of vegetation (where, presumably, you do not get soil moisture retrievals) on your results? You do get some significance extending down to 8N, where it is quite vegetated.

P9, L23-28: While the explanation offered here sounds plausible, I would be wary of drawing conclusions from a few 'blue' points – as you say, this is likely not statistically significant. Also, are you sure less than 0.1% of points have a positive delta(e)? In the $5°$ domain there is less than 100 points (and one 'blue' point).

P12, L23-26: have you looked at the seasonal variability? I wonder if june (or years which are particularly dry) behave differently.

P13, L25: I imagine the point here is that boundary layer moisture in the north is less tied to (local) soil moisture, and depends more on the larger scales (i.e. monsoon intrusions), which is why you don't see a wet advantage even though moister conditions do increase rainfall.

P13, L30: might be worth mentioning that in some cases soil moisture gradients can determine the location of convection, even if the trigger is provided by cold pools or the larger scale (Birch et al 2013) Birch, C. E., D. J. Parker, A. O'Leary, C. M. Taylor, J. H. Marsham, P. Harris, G. Lister, 2013: The impact of soil moisture and atmospheric waves on the development of a mesoscale convective system: A model study of an observed AMMA case, Q. J. R. Meteorol. Soc., 139, 1712-1730, doi:10.1002/qj.2062

P14, L13-24: I find this whole section quite speculative, and as I say in the major comments, doesn't really address the differences in Sahel with the rest of the globe (what about tropical areas? Or other semiarid ones that are different?). Also, I'm not sure I agree with "the above relationship is consistent with the negative spatial but positive temporal SMPC". I can see the link with the positive temporal relationship, but why a negative spatial one?

P15, L1: what's the explanation for this conclusion on predictability?

P15, L3-4: 'supports the relevance' I don't understand this sentence. As far as I can tell all of this could be explained just with the spatial relationship.

P15, L7: wouldn't a positive temporal and negative spatial relationship maximise the moisture flux?

P15, L29-32: I don't understand this point. The reason for filtering water bodies and topography is to isolate the role of soil moisture, because it is very likely that water/mountains are much stronger controls.
Figures

Figure 1: I don't understand how/why you regrid the wind data to a finer grid. In any case, you only plot the streamlines for every ∼5°, so this seems redundant. I suggest you delete the last line. L2: change 'indicates' to 'shows'. L3: state the longitudes for the zonal mean. Change 'rectangular' to 'rectangle'.

Figure 3: the 'golden shading' is very hard to see. I suggest you replace it with stippling. Also, is it necessary for panel a for the contours to go up to 3000? A smaller range would highlight more detail.

Figure 4: 'rectangular' to 'rectangle'.

Figure 5: This plot is not very clear, as it's very hard to match specific runs to the symbol (as there are so many, and they are so small). My suggestion would be to move this information into a table. Potentially, you could also include a box and whisker plot, with one box and whisker for T12, one for G15, a dot for your study, and potentially a dot for T12 TRMM/merged and G15 TRMM/GLEAM (as these are the closest set of observations to what you use). I think this would give a better overview of how your results compare with the literature.

Figure 6: see my comments on the wetlands above regarding panel a.

Figure 9: it is not clear from the caption what is the difference between panels a and b?

Figure 11: provide a bit more detail on what you are showing in the caption (e.g. Se). x axis is time, not daily rainfall (as far as I can tell), and you should give some measure of what timescales you are showing. Y axis is both soil moisture and rainfall. 'rainfall sums' over what period? In the bottom panel, are these many short rainfall events, or persistent rainfall (it looks like the former as it is presented)?.

Figure 12: I like the idea of including a conceptual diagram, but at the moment I don't really follow its logic (particularly the drawing on the left). It doesn't really explain the

[Figure]

coexistence of the two mechanisms either; in the second step ('soil dries out in A'), presumably you return back to where you were in step 1 before it rains (it won't get drier than when it is dry), so why do you get 'stronger than usual SM gradient'?

Editorial

P2, L15. Delete 'recognized'.

P2, L18: change 'subtle' to 'less clear'.

P3, L4: 'anomaly' to 'anomalies'

P4, L3: change 'inset rectangular' to 'dashed rectangle'.

P4, L25: 'into' to 'in'

P4, L27: 'few studies HAVE'.

P5, L1: 'still JUST A few centimetres'

P9, footnote: 'statistics' to 'statistic' (or 'is' to 'are')

P9, L1-2: 'does not exclude or even favour higher' to 'is not expected to affect the' (if I understand it correctly).

P9, L9: 'coherence' to 'agreement'.

P11, L23: 'AN additional area'

P11, L30: 'increase in THE amount'

---

## Referee Comment (RC2) · B.P. Guillod (Referee) · 8 Nov 2017

**General comments**

This paper describes a detailed analysis of soil moisture-precipitation coupling over North Africa. Building upon the work from Taylor et al. (2012) and Guillod et al. (2015), the authors conduct an analysis at a higher resolution which allow them to identify the driving mechanisms in more details than these previous global studies. Among others, they highlight the role of wetlands and irrigated areas, and also study mesoscale convective systems (MCS) (both their impact on the statistical analyses and the impact of soil moisture on these systems).

[Figure]

The manuscript presents a useful study that deserves publication in HESS. It is overall well written, clear and concise, with a few exceptions that deserve improvements listed below. Most of my comments below are minor but there is a number of them, hence I recommend major revisions although they should not be difficult to account for.

I have also listed below a number of typos or edits (e.g. removal of commas). Being myself not a native english speaker, the authors can feel free not to implement these if they are confident that their version is more correct.

I am also happy to forgo anonymity.

Benoit Guillod

**Specific comments** (given as PX,LY for page X, line Y):

- P3,L5-7,L16,L25-26: The mention of the 5 degrees resolution of T12 and G15 analyses (and 1 degree in this study) is somewhat misleading. All three studies analyse events at 0.25 degree, and subsequently aggregated their statistics to 5 degrees boxes (or 1 degree in your case). Please make this clearer at these lines to avoid confusion for readers who are not very familiar with those previous studies.

- P6, L7-9: The event identification and spatial metric (point i) is from T12 but the temporal metric (point ii) is from G15.

- P6,L29-30: "a negative rainfall gradient between Lmax and its adjacent four pixels must be present". I do not understand what the authors mean: if Lmax is the pixel where is rained most, isn't a negative gradient with the neighbouring pixels already ensured? Or perhaps I misunderstand what is meant here, in which case some clarification would be useful.

- P8,L20-22. "As in G15, the weakest negative coupling signal in the Sahelian domain is obtained with the PERSIANN (Precipitation Estimation from Remotely Sensed Information using Artificial Neural Networks) data set (Hsu et al., 1997)." I do not fully understand this sentence since the authors did not use PERSIANN. Do you mean perhaps not "As in G15" but rather "The PERSIANN estimates from G15 exhibit weakest negative spatial coupling from all..." or something along these lines?

- P8, L24-25: It could be stated that the first part relates to the grey lines on Fig. 5 while the second and last part of the sentences is not shown.

- P10,L2: The chosen range (Q25-1.5×IQR, Q75+1.5×IQR) is somewhat complicated to understand. Why wasn't an easier range such as a percentile (e.g. Q01-Q99) or a fixed distance to the mean (e.g., +/- 2 std deviation) chosen? I understand that the choice restricts the selection to values that are very far from the mean and this might not happen at every pixel, but it is not straightforward to understand.

- P10,L5-10: This result might indicate that the use of the median Delta rather than the mean might be more appropriate, i.e. less affected by those extreme values?

- P12,L7-9: This is encouraging and supports the methodology of T12/G15 which was primarily aimed at detecting newly created systems rather than existing, advected MCS. This might be worth noting.

- P13,L9-19: The description of LCL results confused me initially, because Fig. 10b shows the height in hPa but the authors implicitly refer to the height as a distance above ground, both of which are of opposite sign. Hence I was first confused when reading "A slight increase of the LCL in the South" while Fig. 10b shows negative anomalies. I support the implicit use of height above ground in the text, but I suggest the addition of a short sentence that highlights that increase of the LCL height is shown as a decrease, in red, of LCL in hPa - or something along these lines.

- Section 5.2 (role of rainfall persistence): This section is useful and I like the concept behind Figure 11. However, the authors do not discuss explicitly whether rainfall persistence may partly reflects an effect of the land-surface or whether it only reflects atmospherically-driven persistence (the latter implying that the observed statistical relationship would be due to confounding factors). This is, of course, impossible to disentangle from observations alone and it is out of the scope of this paper to fully address this issue. Nonetheless, I feel that it deserves to be at least briefly discussed here. Numerous papers address this topic (e.g., Salvucci et al., 2002; Guillod et al., 2014; Teuling et al, 2005; Seneviratne et al., 2010).

- Figure 2: This is a very useful diagram.

- Figure 4: "The percentile values lying outside the significance range (10-90

- Figure 5: This figure is slightly confusing, although the content is useful. My understanding is that the upper dots are the fraction of negative SMPC and the lower dots are the fraction of positive SMPC, if that is correct this should be stated clearly. However I would suggest to use another way of displaying these, for example as a bar plot and one colour for positive SMPC, one colour for negative SMPC, both of them shown as values above 0 (technically it is the percentage of grid boxes so it cannot be negative). Also, the mean and ST.DEV are not clearly defined: is this the mean/stdev of all the dataset combinations of T12, G15 and your study? Why not show, for instance with light blue lines, the same for positive SMPC?

- Figure 6: "flood planes" -> "flood plains"? Also, why are there grey boxes? Is this where no extreme value is reached?

- Figure 7: "ERA-Interim temperature and specific humidity profile and surface pressure data" -> "ERA-Interim temperature, specific humidity profile and surface pressure data". Also, "their typical state" is unclear, perhaps replace with "their climatology"?

- Figure 8: This is a nice illustration, but it could be improved. Among others: (i) the X axis is not "[DAILY RAINFALL]" but "[TIME]". (ii) The Y-axis is not only soil moisture but also rainfall. (iii) Rainfall appears twice, once as "rain events" in grey bars and once as a solid black line (rainfall sums). Shouldn't it appear only once? Also, I am not sure why rainfall sums follows a sinusoidal shape here, I would favour the grey bars rather than the solid lines. (iv) More generally the caption should better explain the diagram.

If some of these suggestions do not make sense, it probably points to something being unclear which led to a misunderstanding from myself...

**Technical corrections**

- Page 1, line 2: "1 degree horizontal resolution". This is somewhat confusing as the analysis was done on 0.25 degree but the statistics were aggregated to 1 degree.

- Page 1, line 20: "1 to 3-D" -> "1-D to 3-D"?

- Commas (",") are a little over-used in the manuscript. I suggest the authors to check these, here is a non-exhaustive list of where I think should be removed: P2,L2: "Both, observational", P2,L34: "wet soil, can favour...", P7,L11: "To estimate, whether".

- P2,L12: TMPA is used as an acronym but is defined only later, perhaps refer to section 2.3.

- P3,L4: Add a comma before "respectively"?

- P3,L5: "no attempts were made" -> "no attempt was made"?

- P3,L15: "in North African region" -> "in North Africa"?

- P3,L18: "First we focus on identification" -> "First, we focus on the identification"?

- P3,L23: "inter-relate" -> "relate to each other" or "interact"?

- P4,L3: "inset rectangular" -> "dashed rectangle"?

- P4,L5: "2016) and one of the" -> "2016), and as one of the"?

- P6,L8-9 and P6,L26-P7,L3 and P9,L3 etc...: "-" often appear after (i),(ii) etc which could be removed.

- Title of subsection 4.1: replace "." with ":"?

- P12,L30: "anywhere" -> I think the authors meant "everywhere" (or perhaps "almost everywhere").

- P13,L13: "typical value" is somewhat unclear, perhaps only mention "climatological mean"?

- P13,L13: "would imply presence of a stronger..., which can easier..." -> "would lead to stronger..., which can better..."?

- P13,L21: "is shown" -> "has been shown"?

- P16,L9: "Benoi Guillod" -> "Benoit Guillod"

**References**

- Guillod, B. P. et al. Land-surface controls on afternoon precipitation diagnosed from observational data: uncertainties and confounding factors. Atmos. Chem. Phys. 14, 8343–8367 (2014).

- Salvucci, G. D., Saleem, J. A. Kaufmann, R. Investigating soil moisture feedbacks on precipitation with tests of Granger causality. Adv. Water Resour. 25, 1305–1312 (2002).

- Seneviratne, S. I. et al. Investigating soil moisture-climate interactions in a changing climate: a review. Earth-Sci. Rev. 99, 125–161 (2010).

- Teuling, A. J., Uijlenhoet, R. Troch, P. A. On bimodality in warm season soil moisture observations. Geophys. Res. Lett. 32, 10–13 (2005).

---

## Author Comment (AC1) · 10 Dec 2017

The authors highly appreciate the editor's work in organizing the fast and smooth review procedure. The authors are also grateful to the two reviewers for their overall positive evaluation of our work, for their time and useful, concise comments and suggestions which will certainly help us to improve the quality of our paper. The comments of every reviewer are addressed separately. The authors' response is given below every reviewers' remark. The already implemented corrections/ changes are highlighted in the attached pdf file using latexhdiff tracking tool.

The author's response to the Referee #1

[Figure]

The authors thank anonymous reviewer for a thorough evaluation of the manuscript and especially for the valuable remarks on the structure and clarity. For details, please find our response to every item further below.

Major comments REV#1 1. Presentation: I found the paper a bit disjointed to read. I think there are lots of interesting ideas, but it jumps a bit from one thing to the other. I wonder if the results could be reorganised to make it easier to read. One suggestion is to first describe the feedbacks, i.e. show both the spatial and temporal soil moisture – precipitation coupling (figures 4, 5, 9, maybe 10), and then have a section talking about processes. The description of processes should then be consistent with both the spatial and temporal feedbacks you observe. The language throughout could be improved a bit as well (I have included some suggestions below, but my comments are not comprehensive). Finally, some figures I think could also be improved (my suggested changes are in the minor comments).

AR: We thank the reviewer for his careful evaluation and proposed solutions. After weighing the arguments of the referee carefully, we believe that restructuring the paper does not fit with the structure that we have foreseen. That being said, we realize that we have not clarified the structure well enough. We chose our ordering, because the analyses presented in Sections 4.3, 4.4 and related to a processes part are applied to the spatial SMPC only. The temporal SMPC is originally meant to be a secondary result, and is used here as one of the criteria to approve/ reject consistency to the mechanism of local breeze-like circulations on moist convection development.

REV#1 2. Wetlands (S 4.3): Of the results, this is the part that I found most confusing, and therefore least convincing. Some points: - Your definition of extreme is very confusing, it would be easier if it were expressed simply as the values above/below given percentiles (which is more or less what is done here, as far as I can tell, but with a fixed offset as well).

AR: Following the reviewers' advice, the definition criterion for an extreme value using

varying percentile thresholds have been tested. Unlike the original extreme value definition, application of percentile thresholds will always result in the identified outlier in every grid box due to the way the percentile limits (1st and 99th percentile) are calculated. In that case, we would need to justify somehow an additional offset selection. Differently, originally chosen Q25 – 1.5*IQR and Q75 + 1.5*IQR thresholds on the contrary identify the values that are anomalously "far" from the sample, and hence lead to identification of only outliers and extremes. Therefore, we decided to preserve the original definition of extreme value in the study, since it is also a rather commonly used and justified extreme value definition. Yet, to support the text explanations and to make the approach clearer, an additional schematic was added to the Figure 6.

REV#1 Wetlands (S 4.3): - I agree that features such as wetlands are likely to lead to extreme soil moisture gradients, but you might also expect the temporal variability to be low (at least over the wet part). Given you calculate temporal statistics, why not use this as an additional criterion? i.e. find locations with high spatial and low temporal variability. - The map (fig 6a) is not all that useful, as it is hard to see the exact locations and extent of topographic and wetland regions. It is therefore hard to tell if the conclusions in the last two paragraphs are really substantiated. For example, the region of extremes in the south East is very large, and I can't really tell how much of it falls on actual wetland. Similarly, the Gezira scheme in the map is quite small, and it's quite hard to tell what points on the significance contours you are linking to it.

AR: We agree with the reviewer that the explicit quantitative link from the identified location of extremes to the wetlands is missing, and would add a value to our conclusions. We elaborated on the reviewer's suggestions and some of our earlier tests to get an estimate of locations likely covered by wetlands (See Fig. 6c). Based on this analyses we will modify the paragraph in Section 4.3 and Fig 6 accordingly. Interestingly, all of identified areas of extremes including the large region in the East resembles pretty well distribution of wetlands obtained by Matthews&Fung from the pilot observations (see Fig.6b) The exception only comprises the floodplains south of Chad lake for unknown

reason.

The added analysis is based on the linear regression estimates of 1-day soil moisture drying rate (SM at day 0 "minus" SM at day -1) and a starting soil moisture value (i.e. SM at day -1) climatologies. The climatologies are calculated for every Lmin location, (for same month as the event but for the non-event years). As the output, we consider the Lmin locations, where climatological values of the drying rate are always small and do not vary much with the initial soil moisture, as being representative for a water body or a wetland. Finally, 1x1 deg boxes on the Fig 6c, which contain the identified Lmin locations, are marked with a back cross. The detailed explanations will be added into Appendix.

REV#1 - Following on from this, in Fig 6b the dots (where the median and mean are opposite signs) do not really match the extremes (colors). I'm therefore not sure if the statement is justified that "extreme [soil moisture gradients] . . . in some cases appear to predefine its significance".

AR: We modified a bit the wording in the paragraph, so we hope that the link between the extremes and significance of the coupling is clearer now (P9-L13). Generally, the extremes will predefine the significance of 1x1 deg boxes not because of their effect on the sign change but due to their influence on the sample mean. The change in the sample mean in turn will affect the magnitude of the departure from the control (climatology), and hence the coupling significance. That is why simple removal of extreme values from the samples leads to a 30% reduction in the amount of 1x1 deg boxes with significant SMPC (P10-L8). The sign change is rather considered here as another conclusion stating that in most of the cases presence of extremes in a sample does not affect SMPC parameter sign (difference between median and mean).

REV#1 - More mechanistically, could the wetlands not be partly the cause of the covariability between the spatial and temporal feedbacks? The wetlands are approximately spatially and temporally constant, therefore when the soils surrounding it are drier than

normal, this will always represent both a temporal and spatial negative soil moisture anomaly.

AR: It is very hard to answer this question with some degree of certainty for a number of reasons: 1) If the wetland location is always wet, then a presence of negative gradient is guaranteed. Yet the latter does not exclude the possibility of a drier location to be represented by a small positive SM anomaly, i.e. a positive temporal SMPC. 2) correlation of temporal and spatial coupling does not reveal particular signature over wetlands (Fig 9c). It is more likely that the cause of the co-variability is primarily a negative gradient itself.

REV#1 3. Reasons for co-variability of spatial and temporal SMPC: The discussion here is again a bit muddled. In the conclusions you state "the drying of the soil for several days . . . may play a role in the opposite sign of the temporal coupling as compared to the positive relationship in wetter climates". It is worth noting that in G15 the temporal coupling is positive across most of the globe, including in other arid and semiarid regions (northern India, Australia, Saudi Arabia, etc.). The 3- 4 day variability of rainfall in West Africa driven by African easterly waves is a factor, as is pointed out. I think, however, that the primary factors is probably that this is a high CIN/high CAPE environment, therefore anything that helps overcome the CIN will enhance rainfall. This is mentioned in the text when comparing the southern and northern part of the domain, but I suspect it can also explain the differences with other areas of the globe.

AR: We thank the reviewer for being objectively critical. 1) We may not fully agree on the first point stating that ". . . in G15 the temporal coupling is positive . . . including in other arid and semiarid regions". From their suppl. Fig.3 it is seen that the semi-arid regions like the Great Plains and the Sahel have expressed negative temporal coupling. Most of the positive temporal SMPC weirdly is identified in the deserts, inc. Saudi Arabia, S. Africa and Australia. Australia, inter alia, was not identified as a SMPC "hot-spot" in the studies of Koster et al., 2004 and Dirmeyer et al., 2011. Northern India might still express an orography effect. 2) It is indeed worth mentioning the relevance

of BL recovery linked with building up CAPE and depleting CIN. Thank you for the comment. Yet, to our perception main point here is rather in the effect of cyclicity of rain systems and soil drying periods on the temporal coupling sign in the Sahelian semi-arid environment verses role of synoptic system and rainfall persistence in wetter climates.

Minor comments REV#1 P2, L8: 'have a direct effect on the resulting sign'. It would be useful to explicitly state what the sign they find is (i.e. positive temporal, negative spatial) AR: The sentence was reformulated.

REV#1 P4, L9-10: 'on the northern flank. . .' In the diagram it looks like the gradients go across the ITCZ, as opposed to just being on the northern side? AR: The sentence was reformulated.

REV#1 P6, L29-30. You say you need at least three values of Lmin – do you take the three lowest? (presumably, unless Lmin is 0, there is only one minimum value). I am also confused by the criterion that 'negative rainfall gradient between Lmax and its adjacent four pixels must be present'. If Lmax is the maximum, then the gradient with the 4 pixels surrounding it will always be negative – I suspect I am misunderstanding this line.

AR: 1) Indeed, in most cases if Lmin is non-zero, then it is likely that there will be only one Lmin value around given Lmax. Hence, most often values of aft. accum. rainfall in Lmin locations will be 0.0 mm. The clarification sentence was added in the text following your remark (P6-L30). 2) It is indeed a confusing sentence. Thank you for pointing it out. In fact, identification of a local maximum does not automatically exclude the chance of having similar cum. rainfall value in a neighboring pixel. Minima locations are not necessarily the neighbors of Lmax. Therefore an additional criterion is required to proof that Lmax is an absolute maximum within a box. As it was stated in the following sentence (P6-L30/31), such a criterion also helps to eliminate number of events identified within or at the edge of squall-lines. Following your remark, we

decided to exclude this sentence from the paper, as it is rather a technical detail, and does not add much to understanding of the results.

REV#1 P8, L9: '. . .orography mask applied in this study'. Do you mean the maxima are produced by the fact that you are masking the region around the maxima? This sentence is not very clear. AR: The sentence was reformulated.

REV#1 P8, L31: I wonder if the different datasets agree more also because precipitation retrievals themselves are more consistent over flat terrain, while they are likely to disagree more over complex topography.

AR: In general, the areas of strong geographical gradients are masked out in the study. This however would not necessarily mean that we are left with flat terrain only. The (dis-)agreement between experiments is surely a combination of uncertainties coming from both, soil moisture and rainfall data sets. More complex terrain and hence more recurrently flooded areas towards East are expected generally to complicate both, the accuracy of soil moisture and rainfall data sets, as well as ability to isolate surface effects on rainfall. It is therefore likely that orography influences the results. Yet, at which degree and if it could be a dominant factor for a (dis-) agreement between the experiments is hard to answer without carrying out more analysis.

REV#1 P9, L19: might be worth mentioning that the significant correlations lie exactly on the semiarid transition zone between forest (in the south) and the desert. Also, have you considered the impact of vegetation (where, presumably, you do not get soil moisture retrievals) on your results? You do get some significance extending down to 8N, where it is quite vegetated.

AR: 1) We deliberately decided to not mention the link to the land cover or the transition (Sahel) zone. Mainly because the high correlations partly reach quite far south as you mention, but also because the high correlations appear as a dipole rather than a zonal feature. 2) By method, the soil moisture pixels are excluded if vegetation optical depth goes beyond 0.8. It is a common threshold that is usually used to filter out effect

of vegetation on soil moisture quality. Effect of vegetation on the SMPC results was not assessed within the current framework. It is expected however that vegetation (especially in the south and during wettest months of July-August) will influence SMPC relationship via its effect on turbulent flux partitioning. In this way, application of soil moisture parameter instead of e.g. surface fluxes is a definite drawback of the given method, which can be explored in the future studies.

REV#1 P9, L23-28: While the explanation offered here sounds plausible, I would be wary of drawing conclusions from a few 'blue' points – as you say, this is likely not statistically significant. Also, are you sure less than 0.1% of points have a positive delta(e)? In the 5 ẘ domain there is less than 100 points (and one 'blue' point).

AR: 1) In the lines (now L28-33) the link between the "blue" points and location characteristics is rather meant to be a hypothesis to prove in the following section 4.3 (extremes). Then, in the section 4.3 as well as in the newly added analyses on wetland locations we actually illustrate that extreme positive soil moisture gradients indeed emerge in this concrete location and coincide with wetland positioning. As a potential solution we could suggest adding a sentence after the L28-33: "... The potential link between the land surface characteristics and SMPC parameter will be explored in more detail in the following section 4.3. " 2) Thank you very much for checking on the numbers. We seem to have forgotten that the table values are in % for these few experiments. The values have been corrected now.

REV#1 P12, L23-26: have you looked at the seasonal variability? I wonder if June (or years which are particularly dry) behave differently.

AR: We did check very briefly if the sensitivity of the SMPC signal to the choice of summer month is consistent with the result presented in the study of Taylor et al., 2011 (suppl. FigS4). However, in order to preserve maximum sample size, all further calculations were carried out for JJAS months jointly.

REV#1 P13, L25: I imagine the point here is that boundary layer moisture in the north is

less tied to (local) soil moisture, and depends more on the larger scales (i.e. monsoon intrusions), which is why you don't see a wet advantage even though moister conditions do increase rainfall.

AR: Indeed, that is also how we understand the result.

REV#1 P13, L30: might be worth mentioning that in some cases soil moisture gradients can determine the location of convection, even if the trigger is provided by cold pools or the larger scale (Birch et al 2013)

AR: The remark has a good point, but in our opinion it falls out a bit of the context of the paragraph if added (now P13, L22). The paragraph and the result indicate that the mean number of 10 dry days in the northern latitudes (>15N) will unlikely lead to any strong soil moisture heterogeneity. Therefore other triggering processes in combination with increased moisture advection will likely favour moist convection development. In this sense our result would be more consistent with the listed studies of Barthe et al., 2010 and Cuesta et al., 2010. The MCS case study of Birch et al,. 2013 lies exactly at the boarder (~15N) of the increased BL moisture pattern in our Figure 10b.

REV#1 P14, L13-24: I find this whole section quite speculative, and as I say in the major comments, doesn't really address the differences in Sahel with the rest of the globe (what about tropical areas? Or other semiarid ones that are different?). Also, I'm not sure I agree with "the above relationship is consistent with the negative spatial but positive temporal SMPC". I can see the link with the positive temporal relationship, but why a negative spatial one?

AR: We agree with the reviewer's remark on the link to spatial coupling. The negative spatial coupling was probably mentioned there by a mistake. We also plan to elaborate more on the section 5.2 (rainfall persistence) and place it out of the main results story line to a discussion section.

REV#1 P15, L1: what's the explanation for this conclusion on predictability?

AR: The assumption on predictability is based on the identified negative spatial SMPC and on the conclusions made in the Section 5.2. The positive temporal SMPC in wet climates is likely to reflect rainfall persistence linked to persistence in synoptic situations. Hence, it can be expected to provide some predictability to rainfall. In the semi-arid African region, negative temporal coupling is tightly linked to the drying of the soil in time. Therefore, it is unlikely that temporal SMPC alone provides any information on the future rainfall as the soils experience drying cycle all the time. Yet, the identified significant negative spatial SMPC hints on a possibility that next rain will happen in the vicinity of the previous, therefore providing some predictability potential for rain.

REV#1 P15, L3-4: 'supports the relevance' I don't understand this sentence. As far as I can tell all of this could be explained just with the spatial relationship.

AR: Indeed, the spatial relationship alone would be consistent with the mechanism of "breeze-like" circulations. Yet, in combination with a positive temporal coupling in the conditions of the Sahel, formation of local circulations in our understanding would be less likely. Positive temporal coupling in the Sahel environment typically means that it rained 1, max 2 days ago, and the soil (in Lmax location) is wet. Hence, a smaller spatial negative gradients in soil moisture and a higher (lower) moisture (heat) flux can be expected. This altogether would theoretically lead to an additional cooling, less vigorous updrafts, and hence a decreased likelihood to form thermal rolls. The combination with the temporally drier soils (i.e. negative temporal SMPC) in Lmax location is on the contrary expected to result in a higher buoyancy flux, stronger spatial gradients and hence facilitate likelihood of breeze-like circulations. In general, the reviewer's remark is a valid and an open research question, which can be addressed in the future. Following existing model experiments (Avissar&Schmidt 1998), higher mean sensible heat flux conditions would also require stronger spatial gradients to form thermal rolls.

REV#1 P15, L7: wouldn't a positive temporal and negative spatial relationship maximize the moisture flux?

AR: We would not think it would be the case for the studied semi-arid region because of the argumentations brought above. For wetter latitudes, the results from the paper of e.g. Taylor et al., 2015 also support higher likelihood of "breeze-like" circulation mechanism in the conditions of less antecedent rainfall so that the soil moisture limited regime can be archived.

REV#1 P15, L29-32: I don't understand this point. The reason for filtering water bodies and topography is to isolate the role of soil moisture, because it is very likely that water/mountains are much stronger controls.

AR: It seems that the sentence was not well formulated as it caused confusion. We have edited it, and removed the reference to orography. Overall, main point here was about the gaps in the filtering of water bodies. The water body mask in the present method is static and does not take into account variability in flood plains between and within years. That is also the main reason why we identify a prominent link of the rainfall and extreme soil moisture gradients to the location of wetlands. In the future, application of existing dynamical wetland products like the one from C. Prigent (https://lerma.obspm.fr/spip.php?article91&lang=en) may be used to eliminate the effect of water bodies on moist convection development.

Figures REV#1 Figure 1: I don't understand how/why you regrid the wind data to a finer grid. In any case, you only plot the streamlines for every âĹij5 âŮ̧ , so this seems redundant. I suggest you delete the last line. L2: change 'indicates' to 'shows'. -L3: state the longitudes for the zonal mean. Change 'rectangular' to 'rectangle'.

AR: ERA-Interim wind data was re-gridded using bilinear interpolation method of the CDOs (Climate Data Operators) tool to keep consistency to observational data sets applied in the study. We agree with the reviewer, that for the purpose of this plot the re-griding step could have been omitted. As follows, the above suggested wording changes have been implemented.

REV#1 Figure 3: the 'golden shading' is very hard to see. I suggest you replace it with
stippling. Also, is it necessary for panel a for the contours to go up to 3000? A smaller range would highlight more detail.

AR: The figures were replotted and a mask presentation was improved.

REV#1 Figure 4: 'rectangular' to 'rectangle'.

AR: The word was corrected everywhere throughout the text. Thank you for the careful review.

REV#1 Figure 5: This plot is not very clear, as it's very hard to match specific runs to the symbol (as there are so many, and they are so small). My suggestion would be to move this information into a table. Potentially, you could also include a box and whisker plot, with one box and whisker for T12, one for G15, a dot for your study, and potentially a dot for T12 TRMM/merged and G15 TRMM/GLEAM (as these are the closest set of observations to what you use). I think this would give a better overview of how your results compare with the literature.

AR: Following both reviewers suggestions, the figure was replotted in a more clear manner, as well as the data from the figure was additionally summarized in the Table A1. The Table A1 was placed to the appendix section for the moment.

REV#1 Figure 6: see my comments on the wetlands above regarding panel a.

AR: Additional analyses will be done to identify potential wetland locations (see in the above comments on wetlands), and the Fig 6b presenting an extreme gradients will be updated and improved. The Fig 6a will be removed.

REV#1 Figure 9: it is not clear from the caption what is the difference between panels a and b?

AR: The caption text was changed and is hopefully more clear now. In general, the panel (a) shows soil moisture anomaly values averaged over 1x1 deg boxes, while panel (b) indicates the departure of these averaged anomaly values from their typical

(non-event) conditions.

REV#1 Figure 11: provide a bit more detail on what you are showing in the caption (e.g. Se). x axis is time, not daily rainfall (as far as I can tell), and you should give some measure of what timescales you are showing. Y axis is both soil moisture and rainfall. 'rainfall sums' over what period? In the bottom panel, are these many short rainfall events, or persistent rainfall (it looks like the former as it is presented)?.

AR: The figure will be modified following the reviewer's comments

REV#1 Figure 12: I like the idea of including a conceptual diagram, but at the moment I don't really follow its logic (particularly the drawing on the left). It doesn't really explain the coexistence of the two mechanisms either; in the second step ('soil dries out in A'), presumably you return back to where you were in step 1 before it rains (it won't get drier than when it is dry), so why do you get 'stronger than usual SM gradient'?

AR: We thank the reviewer for questioning some of the proposed concepts. The figure was improved now, and it will additionally be elaborated following the modification to the Section 5.2 (persistence).

Editorial REV#1 P2, L15. Delete 'recognized'. P2, L18: change 'subtle' to 'less clear'. P3, L4: 'anomaly' to 'anomalies' P4, L3: change 'inset rectangular' to 'dashed rectangle'. P4, L25: 'into' to 'in' P4, L27: 'few studies HAVE'. P5, L1: 'still JUST A few centimetres' P9, footnote: 'statistics' to 'statistic' (or 'is' to 'are') P9, L1-2: 'does not exclude or even favour higher' to 'is not expected to affect the' (if I understand it correctly). P9, L9: 'coherence' to 'agreement'. P11, L23: 'AN additional area' P11, L30: 'increase in THE amount. AR: All correction were implemented.

Please also note the supplement to this comment:
https://www.hydrol-earth-syst-sci-discuss.net/hess-2017-530/hess-2017-530-AC1-supplement.pdf

**Supplement:**

[revised manuscript text omitted]

**Figure 6.** (a) -  Scematic box plot illustrating the ($Q_{25}$-1.5 ×$IQR$, $Q_{75}$+1.5 ×$IQR$) range used to identifiy extreme $\Delta(S_e'^{Lmax})$ values. Here, $Q_{75}$ and  $Q_{25}$ are the third and first quartiles respectively, and the interquartile range (IQR)  is the  difference between them. (b) - Natural wetland fraction from Matthews and Fung1987 on a $1°$ × $1°$ grid (adopted from Fig 3 in Prigent et al, 2007). (c) - Distribution of soil moisture gradient $\Delta(S_e'^{Lmax})$ extremes in the corresponding event sample of a $1°$ × $1°$ box (color). $\Delta(S_e'^{Lmax})$ is considered to be an extreme if it lies outside the ($Q_{25}$-1.5 ×$IQR$, $Q_{75}$+1.5 ×$IQR$) range Black dots indicate boxes, in which $\Delta(S_e'^{Lmax})$ sample mean and median have opposite signs. Black crosses indicate boxes containing $Lmin$ locations, in which climatology of daily soil moisture drying rates does not vary much with different soil moisture conditions. These relationship is equivalet to low soil moiture variability in time, and is representative for a wet (flooded) locations. For detailed algorthym the reader is referred to the Appendix Figure A1. The distribution of identifed potentially flooded locations is consistent with the natural wetland fraction (b).

[revised manuscript text omitted]

---

## Author Comment (AC2) · 10 Dec 2017

The authors highly appreciate the editor's work in organizing the fast and smooth review procedure. The authors are also grateful to the two reviewers for their overall positive evaluation of our work, for their time and useful, concise comments and suggestions which will certainly help us to improve the quality of our paper. The comments of every reviewer are addressed separately. The authors' response is given below every reviewers' remark. The already implemented corrections/ changes are highlighted in the attached pdf file using latexhdiff tracking tool.

The author's response to the Referee #2: Benoit Guillod ————————————————

[Figure]

———————————————-

The authors thank Benoit Guillod for his willingness to review our manuscript, for the detailed assessments and valuable suggestions. We also appreciate Benoit's kind decision to forgo anonymity. It is especially relevant for us to get an evaluation of the manuscript by Benoit, since our work among others is built upon the paper of B. Guillod and his co-authors from 2015.

General comments REV#2 - This paper describes a detailed analysis of soil moisture-precipitation coupling over North Africa. Building upon the work from Taylor et al. (2012) and Guillod et al. (2015), the authors conduct an analysis at a higher resolution which allow them to identify the driving mechanisms in more details than these previous global studies. Among others, they highlight the role of wetlands and irrigated areas, and also study mesoscale convective systems (MCS) (both their impact on the statistical analyses and the impact of soil moisture on these systems). The manuscript presents a useful study that deserves publication in HESS. It is overall well written, clear and concise, with a few exceptions that deserve improvements listed below. Most of my comments below are minor but there is a number of them, hence I recommend major revisions although they should not be difficult to account for. I have also listed below a number of typos or edits (e.g. removal of commas). Being myself not a native english speaker, the authors can feel free not to implement these if they are confident that their version is more correct. I am also happy to forgo anonymity. Benoit Guillod

Specific comments (given as PX,LY for page X, line Y): REV#2 - - P3,L5-7,L16,L25-26: The mention of the 5 degrees resolution of T12 and G15 analyses (and 1 degree in this study) is somewhat misleading. All three studies analyse events at 0.25 degree, and subsequently aggregated their statistics to 5 degrees boxes (or 1 degree in your case). Please make this clearer at these lines to avoid confusion for readers who are not very familiar with those previous studies.

AR: Thank you for bringing this point out. We were aware the resolution verses aggregation may cause confusion. We replaced the word resolution to either horizontal grid or scale consistently throughout the text. The resolution is only referred to data sets. We hope this can solve the confusion.

REV#2 - - P6, L7-9: The event identification and spatial metric (point i) is from T12 but the temporal metric (point ii) is from G15.

AR: The sentence was corrected following the remark.

REV#2 - - P6, L29-30: "a negative rainfall gradient between Lmax and its adjacent four pixels must be present". I do not understand what the authors mean: if Lmax is the pixel where is rained most, isn't a negative gradient with the neighboring pixels already ensured? Or perhaps I misunderstand what is meant here, in which case some clarification would be useful.

AR: Thank you for pointing it out. In fact, identification of a local maximum does not exclude the chance of having similar cum. rainfall value in a neighboring pixel. Minima locations are not necessarily the neighbors of Lmax. Therefore, an additional criterion is required to proof that Lmax is an absolute maximum within a box. As it is stated in the following sentence (P6-L30/31), such a criterion also helps to eliminate number of events identified within or at the edge of squall-lines. Following the reviewers' remark, it was decided to exclude this sentence from the paper, as it is rather a technical detail, and does not add much to the understanding of the results.

REV#2 - - P8,L20-22. "As in G15, the weakest negative coupling signal in the Sahelian domain is obtained with the PERSIANN (Precipitation Estimation from Remotely Sensed Information using Artificial Neural Networks) data set (Hsu et al., 1997)." I do not fully understand this sentence since the authors did not use PERSIANN. Do you mean perhaps not "As in G15" but rather "The PERSIANN estimates from G15 exhibit weakest negative spatial coupling from all..." or something along these lines?

AR: Indeed, your interpretation is correct. We modified the sentence following your

suggestion.

REV#2 - - P8, L24-25: It could be stated that the first part relates to the grey lines on Fig. 5 while the second and last part of the sentences is not shown.

AR: The Figure and the text were corrected.

REV#2 - - P10,L2: The chosen range (Q25-1.5×IQR, Q75+1.5×IQR) is somewhat complicated to understand. Why wasn't an easier range such as a percentile (e.g. Q01-Q99) or a fixed distance to the mean (e.g., +/- 2 std deviation) chosen? I understand that the choice restricts the selection to values that are very far from the mean and this might not happen at every pixel, but it is not straightforward to understand.

AR: Following the reviewers' advice, the definition criterion for an extreme value using varying percentile thresholds have been tested. Unlike the original extreme value definition, application of percentile thresholds will always result in the identified outlier in every grid box due to the way the percentile limits (1st and 99th percentile) are calculated. In that case, we would need to justify somehow an additional offset selection. Differently, originally chosen Q25 – 1.5*IQR and Q75 + 1.5*IQR thresholds on the contrary identify the values that are anomalously "far" from the sample, and hence lead to identification of only outliers and extremes. Therefore, we decided to preserve the original definition of extreme value in the study, since it is also a rather commonly used and justified extreme value definition. Yet, to support the text explanations and to make the approach clearer, an additional schematic was added to the Figure 6.

REV#2 - - P10,L5-10: This result might indicate that the use of the median Delta rather than the mean might be more appropriate, i.e. less affected by those extreme values?

AR: Indeed, it is so. Using median instead would reduce the magnitude of delta and hence, the amount of significant boxes, though the spatial pattern will remain the same.

REV#2 - - P12,L7-9: This is encouraging and supports the methodology of T12/G15 which was primarily aimed at detecting newly created systems rather than existing,

advected MCS. This might be worth noting.

AR: It is a good idea, but it seems that the sentence was a bit misleading. We did see that the majority of strong negative gradients is attributed to the first rainfall at the earliest time step, but we did not further analyze weather these rain systems were formed locally. Hence, we would be careful making a statement on the nature of rain systems. The sentence was reformulated a bit to avoid the confusion.

REV#2 - - P13,L9-19: The description of LCL results confused me initially, because Fig. 10b shows the height in hPa but the authors implicitly refer to the height as a distance above ground, both of which are of opposite sign. Hence I was first confused when reading "A slight increase of the LCL in the South" while Fig. 10b shows negative anomalies. I support the implicit use of height above ground in the text, but I suggest the addition of a short sentence that highlights that increase of the LCL height is shown as a decrease, in red, of LCL in hPa - or something along these lines.

AR: Thank you for the careful evaluation. Indeed, it reads confusing. We modified the text now following your suggestions.

REV#2 - - Section 5.2 (role of rainfall persistence): This section is useful and I like the concept behind Figure 11. However, the authors do not discuss explicitly whether rainfall persistence may partly reflects an effect of the land-surface or whether it only reflects atmospherically-driven persistence (the latter implying that the observed statistical relationship would be due to confounding factors). This is, of course, impossible to disentangle from observations alone and it is out of the scope of this paper to fully address this issue. Nonetheless, I feel that it deserves to be at least briefly discussed here. Numerous papers address this topic (e.g., Salvucci et al., 2002; Guillod et al., 2014; Teuling et al, 2005; Seneviratne et al., 2010).

AR: Thank you for pointing this important difference out. We plan to elaborate on the section 5.2, and following your remark we will add a brief clarification on the nature of persistence and references to the discussion text.

Figures REV#2 - - Figure 2: This is a very useful diagram.

AR: Thank you for sharing a positive comment.

REV#2 - - Figure 4: "The percentile values lying outside the significance range (10-90

AR: The sentence was rephrased.

REV#2 - - Figure 5: This figure is slightly confusing, although the content is useful. My understanding is that the upper dots are the fraction of negative SMPC and the lower dots are the fraction of positive SMPC, if that is correct this should be stated clearly. However I would suggest to use another way of displaying these, for example as a bar plot and one colour for positive SMPC, one colour for negative SMPC, both of them shown as values above 0 (technically it is the percentage of grid boxes so it cannot be negative). Also, the mean and ST.DEV are not clearly defined: is this the mean/stdev of all the dataset combinations of T12, G15 and your study? Why not show, for instance with light blue lines, the same for positive SMPC?

AR: We thank the reviewer for his suggestions. The figure was replotted accordingly, and hopefully looks much simper and clearer now. The data from the figure was additionally summarized in the Table A1, which was placed in the appendix section for the moment.

REV#2 - - Figure 6: "flood planes" -> "flood plains"? Also, why are there grey boxes? Is this where no extreme value is reached?

AR: Indeed, the grey boxes are indicative for all the other grid boxes, where no extreme values was identified.

REV#2 - - Figure 7: "ERA-Interim temperature and specific humidity profile and surface pressure data" -> "ERA-Interim temperature, specific humidity profile and surface pressure data". Also, "their typical state" is unclear, perhaps replace with "their climatology"?

AR: The suggestions were implemented. Thank you.

REV#2 - - Figure 8: This is a nice illustration, but it could be improved. Among others: (i) the X axis is not "[DAILY RAINFALL]" but "[TIME]". (ii) The Y-axis is not only soil moisture but also rainfall. (iii) Rainfall appears twice, once as "rain events" in grey bars and once as a solid black line (rainfall sums). Shouldn't it appear only once? Also, I am not sure why rainfall sums follows a sinusoidal shape here, I would favour the grey bars rather than the solid lines. (iv) More generally the caption should better explain the diagram. If some of these suggestions do not make sense, it probably points to something being unclear which led to a misunderstanding from myself. . .

AR: Thank you for the detailed suggestions. We will reevaluate the complete Section 5.2 following both reviewers' comments first, and then will also elaborate on the Figure 8 and will include your suggestions.

Technical corrections REV#2 - - Page 1, line 2: "1 degree horizontal resolution". This is somewhat confusing as the analysis was done on 0.25 degree but the statistics were aggregated to 1 degree. - Page 1, line 20: "1 to 3-D" -> "1-D to 3-D"? - Commas (",") are a little over-used in the manuscript. I suggest the authors to check these, here is a non-exhaustive list of where I think should be removed: P2,L2: "Both, observational", P2,L34: "wet soil, can favour...", P7,L11: "To estimate, whether". - P2,L12: TMPA is used as an acronym but is defined only later, perhaps refer to section 2.3. - P3,L4: Add a comma before "respectively"? - P3,L5: "no attempts were made" -> "no attempt was made"? - P3,L15: "in North African region" -> "in North Africa"? - P3,L18: "First we focus on identification" -> "First, we focus on the identification"? - P3,L23: "inter-relate" -> "relate to each other" or "interact"? - P4,L3: "inset rectangular" -> "dashed rectangle"? - P4,L5: "2016) and one of the" -> "2016), and as one of the"? - P6,L8-9 and P6,L26-P7,L3 and P9,L3 etc...: "-" often appear after (i),(ii) etc which could be removed. - Title of subsection 4.1: replace "." with ":"? - P12,L30: "anywhere" -> I think the authors meant "everywhere" (or perhaps "almost everywhere").
REV#2 - - P13,L13: "typical value" is somewhat unclear, perhaps only mention "climatological mean"? - P13,L13: "would imply presence of a stronger..., which can easier..." -> "would lead to stronger..., which can better..."? - P13,L21: "is shown" -> "has been shown"? - P16,L9: "Benoi Guillod" -> "Benoit Guillod"

AR: Thank you for the thorough evaluation Benoit and the suggestions. Your comments will be implemented, and an additional check up on the grammar and punctuation will be done.

Please also note the supplement to this comment:
https://www.hydrol-earth-syst-sci-discuss.net/hess-2017-530/hess-2017-530-AC2-supplement.pdf

**Supplement:**

[revised manuscript text omitted]

**Figure 6.** (a) -  Scematic box plot illustrating the ($Q_{25}$-1.5 ×$IQR$, $Q_{75}$+1.5 ×$IQR$) range used to identifiy extreme $\Delta(S_e'^{Lmax})$ values. Here, $Q_{75}$ and  $Q_{25}$ are the third and first quartiles respectively, and the interquartile range (IQR)  is the  difference between them. (b) - Natural wetland fraction from Matthews and Fung1987 on a $1°$ × $1°$ grid (adopted from Fig 3 in Prigent et al, 2007). (c) - Distribution of soil moisture gradient $\Delta(S_e'^{Lmax})$ extremes in the corresponding event sample of a $1°$ × $1°$ box (color). $\Delta(S_e'^{Lmax})$ is considered to be an extreme if it lies outside the ($Q_{25}$-1.5 ×$IQR$, $Q_{75}$+1.5 ×$IQR$) range Black dots indicate boxes, in which $\Delta(S_e'^{Lmax})$ sample mean and median have opposite signs. Black crosses indicate boxes containing $Lmin$ locations, in which climatology of daily soil moisture drying rates does not vary much with different soil moisture conditions. These relationship is equivalet to low soil moiture variability in time, and is representative for a wet (flooded) locations. For detailed algorthym the reader is referred to the Appendix Figure A1. The distribution of identifed potentially flooded locations is consistent with the natural wetland fraction (b).

[revised manuscript text omitted]

---

## Referee Report (RR1)

*Review of "Regional co-variability of spatial and temporal soil moisture-precipitation coupling in North Africa: an observational perspective"*

**General comments**

This study uses both spatial gradients and temporal anomalies to investigate the relationship between soil moisture and precipitation over the Sahel. It is very well-written and well-presented, and I appreciate its focus on a single region and the thorough meteorological and geographical understanding brought to bear.

My primary concern is about the statistical rigor of the significance testing: "Significance is represented by a percentile of the [difference in mean metrics] in a bootstrapped sample." Presumably this is done for each grid point individually, but these grid points are not statistically independent and one is almost guaranteed to have Type I errors (erroneous rejections and overstated results). This statistical dependence *between* spatial and temporal metrics is discussed on page 13, but not acknowledged *within* the individual metrics. The more rigorous manner of constructing Figures 4 , 5, or 9 would be to use field significance as in Wilks 2006 JAMC *"On Field Significance" and the False Discovery Rate*. This would better validate the stated "strong preference for convective rainfall over spatially drier soils" or that "temporally negative SMPC dominates". However, it would require substantial reworking of the manuscript in its current form, so this matter is left to the discretion of the authors and editor. Otherwise I have clarifying questions and suggestions:

**Specific comments**

I understand that the statistical framework being used is described in Taylor et al. 2012, but more clarification would still be helpful in reading through the methods:

- Page 7, Line 3 – Why is the accumulated precipitation threshold prior to the rain event lowered to 0? This seems overly stringent to me.
- Page 7, Lines 12-13 – The climatological mean is subtracted from $S'$ prior to $L_{max}$-$<L_{min}>$ gradient calculation. The entire current year is excluded so that the rain event does not impact this climatological mean, but are other years also excluded if a rain event occurred then?
- Page 7, Lines 15-17 – It is confusing to use $Y_e$ as both the spatial gradient and temporal anomaly. The $S'_e$ ($L_{max}$) is calculated in this temporal anomaly exactly as within the spatial gradient, right? A sentence stating this explicitly might also be helpful.
- Page 7, Lines 19-20 – Again I am curious whether the calculation of $Y_c$ accounts for rain events in years other than the one under consideration.

Page 8, Lines 22 to 24 – Although they do not come from this study, I would have appreciated more explanation of the anomalous positive spatial correlations in Figure 5. The mention of "lower consistency of PERSIANN precipitation and soil moisture variability in time" is not entirely clear.

Page 9, Lines 31 to 32 – From Figure 4, I think these values for positive $\delta_e$ are listed backwards. It seems that <3% are positive in the 5° grid, while 6% are positive in the 2.5° grid.

Figure 6a – I do not think this Figure is necessary, as it simply illustrates a basic statistical concept.

Figure 6b – Could you please add a colorbar or mention what the colors mean within the caption?

Page 10, Lines 14-15 – This sentence is a bit vague. Which "initial hypothesis"?

Page 10, Lines 22-33 – It is not clear to me why this linkage of extreme soil moisture gradients to flooded areas is done only for $L_{min}$ locations. Would one not be even more interested to pinpoint the locations of intense rainfall (i.e. $L_{max}$) where the drying rate and high soil moisture have low correlation?

Page 10, Line 33 – In my opinion, Figure 11 could be referenced in the wetland breeze discussions because it is quite useful to understand what is being proposed here, even if comes later in the manuscript.

Figures 7c and 7d – The Hovmöller diagram is of rain rate in [mm h$^{-1}$] (not accumulated precipitation in [mm]), right? Why are there two diurnal profiles one after another?

Page 11, Lines 26-27 – I am surprised by the finding that MCS are "short-lived and smaller" in the Western domain. This is not in agreement with some past literature. For example, Zipser et al. 2006 BAMS find some of the most intense thunderstorms globally in West Africa, and studies like Jackson et al. 2009 MWR focus exclusively on the high frequency of West African MCS. Is there a reason for the discrepancy?

Page 12, Lines 9-14 – This result suggest to me that the "choice of a later accumulation time than in T12" stated in Section 3.1.1 has a large impact on, for example, what is shown in Figures 4 and 7. It would be good to mention the potential impact of adjusting accumulation time in the discussion of these two figures.

Figure 9 – It might be nice to do one sensitivity test of the robustness of the temporal anomaly to the definition of "pre-event" (i.e. for what period of time preceding the rain event does this mostly negative anomaly field persist?).

Page 15, Lines 11-12 – This explanation is a nice mechanistic way of understanding the results, but should adjust with a different soil drying rate (which varies and sometimes depends on initial moisture from Figure A1). This adjustment could be described briefly.

The final point leads to a more general question I had: it seems to me that the current temporal anomaly definition from a climatological mean may not well account for the fact that certain locations simply have much greater soil moisture variability than others. To me, a kind of "soil moisture z-score" normalized by a variability would make more sense. Could the authors comment on this?

Boreal spring and autumn are the African rainy season, and I suppose that isolating these months from the others should change the fields. Have the authors investigated seasonality at all? Could some discussion be added, even if new Figures are not?

---

## Author Response (AR2)

**The authors' response to the reviewers' comments on** "Regional co-variability of spatial and temporal soil moisture - precipitation coupling in North Africa: an observational perspective" *by Irina Y. Petrova et al.*

The authors thank the two reviewers for their time and valuable comments. We highly appreciate the consent of Benoit Guillod to review our paper again, and we value the editor's work and time. The comments of every reviewer are addressed in sequence. The latest implemented corrections/ changes are highlighted in the attached pdf file using *latexhdiff* tracking tool.

**The author's response to the Referee #1: Benoit Guillod**

The revised manuscript has improved most aspects that were previously raised by the referees. In my view the paper will be suitable for publication after a few minor corrections listed below. It is a good contribution to the topic of soil moisture - precipitation feedback, whereby the more regional focus can also benefit to global studies.

Minor/technical comments:
- Figure 6(c): It is not clear what the "distribution of soil moisture gradient extremes" are. I presume that what is shown (colorbar) is their mean value? Please specify in the caption.
- Page 17, line 4: "The identified in the study regions of the strong SMPC...". This does not make sense and needs rephrasing.
- A few typos and grammatical errors remain and I recommend the authors to carefully read through a last time. Alternatively these typos might be identified by the Copernicus typesetting/proofreading.

We thank Benoit for the careful assessment of the previously implemented corrections, and for his positive evaluation of our work. All the new comments/suggestions were taken into account and implemented into the manuscript text.

**The author's response to the Referee #2:**

We thank the anonymous reviewer for his/her appreciation of our work, careful evaluation of the manuscript and valuable suggestions. Please, find our response to every item further below.

**General comments**
This study uses both spatial gradients and temporal anomalies to investigate the relationship between soil moisture and precipitation over the Sahel. It is very well-written and well-presented, and I appreciate its focus on a single region and the thorough meteorological and geographical understanding brought to bear. My primary concern is about the statistical rigor of the significance testing: "Significance is represented by a percentile of the [difference in mean metrics] in a bootstrapped sample." Presumably this is done for each grid point individually, but these grid points are not statistically independent and one is almost guaranteed to have Type I errors (erroneous rejections and overstated results). This statistical dependence between spatial and temporal metrics is discussed on page 13, but not acknowledged within the individual metrics. The more rigorous manner of constructing Figures 4 , 5, or 9 would be to use field significance as in Wilks 2006 JAMC "On Field Significance" and the False Discovery Rate. This would better validate the stated

"strong preference for convective rainfall over spatially drier soils" or that "temporally negative SMPC dominates". However, it would require substantial reworking of the manuscript in its current form, so this matter is left to the discretion of the authors and editor. Otherwise I have clarifying questions and suggestions:

We thank the reviewer for being critical. However, the issue raised above was not fully clear to us. The reviewer mentions that the estimation of percentile and significance is done for each grid point. In the paper, however, these values are estimated always for an aggregate of rain events in either 5x5 , 2.5x2.5 or 1x1 degree box. Also, the minimum allowed number of events in every box is considered. We hope that this clarification from our side would resolve referees' concern. Otherwise, we would please ask to send us more detailed formulation of the problem.

**Specific comments**

We thank the reviewer for pointing out all of the following issues, and we reply to every of them in sequence:

I understand that the statistical framework being used is described in Taylor et al. 2012, but more clarification would still be helpful in reading through the methods:
Page 7, Line 3 – Why is the accumulated precipitation threshold prior to the rain event lowered to 0? This seems overly stringent to me.

In general, the application of zero threshold is more logical, since we want to make sure that no rain occurs in preceding hours. Original threshold of 1 mm among other things considers the typical error range of precipitation detection in current products. In this way this more relaxed threshold may maximize sample size of identified events if a precipitation product tends to overestimate the drizzle, for example. In case of TMPA data set, setting the threshold to zero does not affect the results.

Page 7, Lines 12-13 – The climatological mean is subtracted from S' prior to Lmax-<Lmin> gradient calculation. The entire current year is excluded so that the rain event does not impact this climatological mean, but are other years also excluded if a rain event occurred then?

We are not sure, we understand the question correctly. In general, the calculation of soil moisture anomaly is done for every single event separate. Hence, the calculation of the climatology of every event excludes the data of the corresponding year in which the event occurred.

Page 7, Lines 15-17 – It is confusing to use Ye as both the spatial gradient and temporal anomaly. The S'e (Lmax) is calculated in this temporal anomaly exactly as within the spatial gradient, right? A sentence stating this explicitly might also be helpful.

We thank the reviewer for pointing this out. We would be in favour to preserve the Ye notation, since we use it in the following paragraph to explain significance calculation as well as in the schematic Figure 2. Yet, we reformulated some sentences to improve the clarity.

Page 7, Lines 19-20 – Again I am curious whether the calculation of Yc accounts for rain events in years other than the one under consideration. The control sample Yc generally contains climatological days where the rain did not occur.

Page 8, Lines 22 to 24 – Although they do not come from this study, I would have appreciated more explanation of the anomalous positive spatial correlations in Figure 5. The mention of "lower

consistency of PERSIANN precipitation and soil moisture variability in time" is not entirely clear.

The results summarized in Figure 5 were obtained based on the visual analyses of the findings of G15. Hence, a detailed analyzed of all locations with positive SMPC were not feasible. Yet, we explored potential reasons for the positive coupling identified in our set up in Fig 4e, which we discussed in L27-30, p9.

Page 9, Lines 31 to 32 – From Figure 4, I think these values for positive δe are listed backwards. It seems that <3% are positive in the 5° grid, while 6% are positive in the 2.5° grid.

Thank you for the careful assessment. The typo was corrected.

Figure 6a – I do not think this Figure is necessary, as it simply illustrates a basic statistical concept.

This figure was added following previous round of the reviews to make the definition of extremes as clear as possible.

Figure 6b – Could you please add a colorbar or mention what the colors mean within the caption?
The colorbar was added following your suggestion. Thank you.

Page 10, Lines 14-15 – This sentence is a bit vague. Which "initial hypothesis"?
The confusing wording was removed. Thank you for pointing at it.

Page 10, Lines 22-33 – It is not clear to me why this linkage of extreme soil moisture gradients to flooded areas is done only for Lmin locations. Would one not be even more interested to pinpoint the locations of intense rainfall (i.e. Lmax) where the drying rate and high soil moisture have low correlation?

The remark has a good point. In general the test for Lmin locations is done in accordance with the "breeze-like" circulation theory. The latter states that local circulation formed between the wetland and its surrounding will cause triggering or re-intensification of rain systems next to the wetland, and its suppression over the wetland (see e.g. Taylor2010). Since the afternoon rainfall maximum is located over Lmax, the Lmin location theoretically should represent the wetland, and therefore location with small soil moisture variability. On the contrary, in Lmax location soil has a chance to dry out, creating a strong (extreme) negative gradient to a flooded region.

Page 10, Line 33 – In my opinion, Figure 11 could be referenced in the wetland breeze discussions because it is quite useful to understand what is being proposed here, even if comes later in the manuscript.
Thank you for the suggestion. We decided to keep it as it is, as more clarifications would be needed in the text, if the reference would be added.

Figures 7c and 7d – The Hovmöller diagram is of rain rate in [mm h-1] (not accumulated precipitation in [mm]), right? Why are there two diurnal profiles one after another?

Indeed, the diurnal cycle is estimated at the original TMPA resolution, i.e. 3-hourly rain rate (mm/hr). The profiles are identical and are duplicated for clarity of their periodicity.

Page 11, Lines 26-27 – I am surprised by the finding that MCS are "short-lived and smaller" in the Western domain. This is not in agreement with some past literature. For example, Zipser et al. 2006

BAMS find some of the most intense thunderstorms globally in West Africa, and studies like Jackson et al. 2009 MWR focus exclusively on the high frequency of West African MCS. Is there a reason for the discrepancy?

Thank you for pointing it out. In the paragraph, we say that the MCS in West Africa are not short-lived, but they are shorter-lived. This is meant in comparison with those typically occurring in the East. In general , this finding is supported and illustrated in the studies of Mathon&Laurant2001 and Laing&Carbone2008;2009. These studies are also references in our manuscript.

Page 12, Lines 9-14 – This result suggest to me that the "choice of a later accumulation time than in T12" stated in Section 3.1.1 has a large impact on, for example, what is shown in Figures 4 and 7. It would be good to mention the potential impact of adjusting accumulation time in the discussion of these two figures.

Indeed, consideration of the complete afternoon time period includes all effects in Figure4. The result of the influence on the SMPC measure is shown and discussed in the Figure 8. That would be equivalent to your suggestion we believe.

Figure 9 – It might be nice to do one sensitivity test of the robustness of the temporal anomaly to the definition of "pre-event" (i.e. for what period of time preceding the rain event does this mostly negative anomaly field persist?).

In their 2015 paper, Guillod et al state that they tested the sensitivity of the temporal SMPC to the soil moisture conditions of the previous day, and it lead to similar results (see in their supplementary material). We did not look into this aspect further.

Page 15, Lines 11-12 – This explanation is a nice mechanistic way of understanding the results, but should adjust with a different soil drying rate (which varies and sometimes depends on initial moisture from Figure A1). This adjustment could be described briefly.
The final point leads to a more general question I had: it seems to me that the current temporal anomaly definition from a climatological mean may not well account for the fact that certain locations simply have much greater soil moisture variability than others. To me, a kind of "soil moisture z-score" normalized by a variability would make more sense. Could the authors comment on this?

The effect of soil moisture variability on the SMPC significance is accounted by the comparison of the event sample to its climatological sample. Yet, this of course does not allow judging on the absolute values of soil moisture and its gradient in general, and the resulting bowen ratio in particular. This also remains being main limitation of the present methodology. The normalization would allow comparison of the e.g. SM  gradient strength between different locations, but will not resolve the issue of a missing bowen ratio.

Boreal spring and autumn are the African rainy season, and I suppose that isolating these months from the others should change the fields. Have the authors investigated seasonality at all? Could some discussion be added, even if new Figures are not?

In the considered semi-arid region, rain season is limited to monsoon JJAS months. The effect of varying monsoon activity stage was briefly considered in work of Petrova2017.
Seasonality per se was tested by Guillod2015 study.

[revised manuscript text omitted]